Introducing LAB60: A 1/60° NEMO 3.6 numerical simulation of the Labrador Sea
Clark Pennelly[1*] and Paul G. Myers[1]
[1]1-26 Earth Sciences Building, University of Alberta, Edmonton, Alberta, Canada, T6G 2E3
*Correspondence to: Clark Pennelly (pennelly@ualberta.ca)
Abstract

7        A high-resolution coupled ocean-sea ice model is set up within the Labrador Sea. With a

horizontal resolution of 1/60°, this simulation is capable of resolving the multitude of eddies
which transport heat and freshwater into the interior of the Labrador Sea. These fluxes strongly
govern the overall stratification, deep convection, restratification, and production of Labrador
Sea Water. Our regional configuration spans the full North Atlantic and Arctic, while the high
resolution is only applied in smaller nested domains within the North Atlantic and Labrador Sea.
Using nesting reduces computational costs and allows for a long simulation from 2002 to the
near-present time. Three passive tracers are also included: Greenland runoff, Labrador Sea
Water produced during convection, and Irminger Water which enters the Labrador Sea along
Greenland. We describe the configuration setup and compare against similarly forced lower-
resolution simulations to better describe how horizontal resolution impacts the representation
of the Labrador Sea in the model.

20       1.  Introduction

21       The Labrador Sea, between Canada and Greenland, plays a crucial role in the climate

system. Situated between the Canadian Arctic and the North Atlantic, multiple current systems
influence this deep basin. Cold and fresh Arctic water flows south through Fram Strait along
Greenland (de Steur et al., 2009), producing the East Greenland Current (EGC). The EGC flows to
the southern tip of Greenland, merging with warm and salty Irminger Water to become the
West Greenland Current (WGC) before flowing northwards along the western coast (Fratantoni
and Pickart, 2007). The WGC flows cyclonically around the Labrador Sea as well as into Baffin
Bay. Significant amounts of freshwater are supplied to this current system from both Davis
(Cuny et al., 2005; Curry et al., 2011; Curry et al., 2014) and Hudson Strait (Straneo and Saucier,
2008) as it travels around the Labrador Sea. The current system is called the Labrador Current
where it merges with the outflow from Hudson Strait (Lazier and Wright, 1993). The Labrador
Current travels southwards along the eastern coast of North America eventually leaving the
Labrador Sea.

Numerous eddies are generated throughout the Labrador Sea, both from high lateral

density gradients which exist during the convection season (Frajka-Williams et al., 2014) as well
as from baroclinic and barotropic instabilities that occur within the boundary currents (Chanut
et al., 2008; Gelderloos et al., 2011). The continental slope along the west coast of Greenland
has a pronounced change in topography that induces instability of the current system,
generating eddies (de Jong et al., 2016). These eddies, known as Irminger Rings, contain a
significant amount of freshwater at the surface as well as subsurface heat. Irminger Rings (15-
30km radius) typically travel southwestwards into the interior of the Labrador Sea and have a
lifespan of up to two years (Lilly et al., 2003). Eddies generated along the Labrador Coast also
contain a significant amount of freshwater (Schmidt and Send, 2007; McGeehan and
Maslowski, 2011; Pennelly et al., 2019). Regardless of where they are produced, these
boundary current eddies often export their properties towards the centre of the basin (Pennelly
et al., 2019), influencing the deep convection which occurs. Convective eddies are generated
from baroclinic instability which arises from large horizontal density gradients during the
convective season (Marshall and Schott, 1999). Convective eddies are much smaller with a
radius between 5 and 18 km (Lilly et al., 2003). These eddies are less studied than the other
eddy types, partly due to a lack of observations (Lilly et al., 2003) as well as their small size
which requires high-resolution models to adequately resolve. Research into the role of each of
the above eddies and their role in restratifying the Labrador Sea is still ongoing; there is no
consensus on which eddy may be more important, though many have narrowed it down to
Irminger Rings and convective eddies (Chanut et al., 2008; Gelderloos et al., 2011; Rieck et al.,

2019).

Deep convection is a rather rare occurrence, only known to occur at a few places in the

ocean. The reason so few places exist is the stringent criteria to produce deep convection: weak
stratification that can be enhanced via isopycnal doming as a result of cyclonic circulation, and
intense air-sea buoyancy loss (Lab Sea Group, 1998; Marshall and Schott, 1999). Cyclonic
circulation and the lateral input of salty Irminger Water helps keep the Labrador Sea weakly
stratified. Furthermore, the Labrador Sea experiences strong heat loss during the winter period
due to the very cold mid-latitude cyclones which frequent the region (Schulze et al., 2016). The
overlying cold and dry air forces a significant flux of heat from the ocean to the atmosphere.
This loss of heat promotes the surface layer to increase in density, overturning the weakly
stratified water column such that the mixed layer can exceed 2000m in depth (Yashayaev,
2007), producing a thick uniform water mass known as Labrador Sea Water (LSW).

Once the convective winter ends, the Labrador Sea quickly restratifies itself within 2-3

months (Lilly et al., 1999), primarily due to large horizontal density gradients that form
convective eddies (Lilly et al., 2003; Rieck et al., 2019) as a result of the deep convection period
(Frajka-Williams et al., 2014). The boundary currents continuously shed eddies with relatively
buoyant water towards the interior Labrador Sea (Straneo, 2006), increasing stratification. This
occurs along the west Greenland and Labrador coasts, though research suggests that the
former supplies more freshwater (Myers, 2005; Schmidt and Send, 2007; McGeehan and
Maslowski, 2011; Pennelly et al., 2019).

LSW is exported out of the Labrador Sea primarily by the Deep Western Boundary

Current (Kieke et al., 2009), though it also spreads eastwards at a slower rate. While LSW is the
lightest component within the Deep Western Boundary Current, it is one of the water masses
which make up the lower limb of the Atlantic Meridional Overturning Circulation (AMOC). As
the overturning circulation transports a significant amount of heat and dissolved gasses
between the equator and polar regions, changes in the production of deepwater can influence
the overturning circulation and ultimately the climate (Bryden et al., 2005). With polar
amplification driven by the positive ice-albedo feedback loop, additional freshwater from
melted ice enters the EGC and WGC (Bamber et al., 2012). The Labrador Sea is experiencing an
increase in freshwater that can be capable of capping convection and preventing LSW from
being formed, ultimately reducing the AMOC strength (Böning et al., 2016). However, a non-
local increase in the surface freshwater flux may promote AMOC strengthening (Cael and
Jansen, 2020) or compensate the local effects of additional freshwater (Latif et al., 2000). Long
climate simulations allow investigation into any AMOC regime shifts that shorter, higher-
resolution simulations may miss. With such different conclusions, freshwater's influence on the
AMOC is not fully known and may vary at different convection regions.

While satellite altimetry provides a wealth of information including sea surface height

anomalies, geostrophic currents, and waves, hydrographic cruises within the Labrador Sea are
often limited to the restratification period when the Labrador Sea is more hospitable for
scientific operations. Argo floats, autonomous drifting profilers which can sample down to
2000m, have become a popular instrument to acquire in-situ data. However, they still lack
coverage within the Labrador Sea which can experience deep convection below their sampling
depth (Yashayaev, 2007). Numerical modelling is a useful tool to explore this data-sparse
region, though it has its limits. Simulations within the Labrador Sea often experience a drift in
model data, producing a Labrador Sea which slowly increases in salinity, and thus density
(Treguier et al., 2005; Rattan et al., 2010). Coarse-resolution simulations suffer even further,
often overproducing the spatial area of deep convection (Courtois et al., 2017), primarily as a
result of not resolving important small-scale features including eddies. These eddies supply the
Labrador Sea with significant heat (Gelderloos et al., 2011) and freshwater fluxes (Hátún et al.,
2007), both strongly impact the stratification, convection, and production of deep water.
Increased horizontal resolution helps produce these eddies and their important fluxes into the
interior of the Labrador Sea but numerical drift still is present within high-resolution
simulations, albeit reduced in severity (Marzocchi et al., 2015).

Numerous high-resolution simulations have been carried out within the North Atlantic.

VIKING20X (Rieck et al., 2019), and its predecessor VIKING20, are global 1/4° simulations which
have a high-resolution 1/20° nest. VIKING20X is a multi-decade simulation which is capable of
resolving eddies within the Labrador Sea. However, simulations with 1/20° horizontal resolution
may not resolve sub-mesoscale processes (Su et al., 2018) that can impact stratification by
carrying heat and freshwater; higher-resolution is needed. The 1/50° HYCOM (Chassignet and
Xu, 2017), 1/60° NATL60 (Fresnay et al., 2018) and eNATL60 (Le Sommer et al., in prep) provide
great insights on the importance of resolving eddies. However, computational expense with
such high-resolution simulations is very high, both in computer time and operational costs. This
often forces higher-resolution simulations to have a reduced length, perhaps only a few years.
The Labrador Sea experiences significant interannual variability (Fischer et al., 2010) and such
short simulations may completely miss any connection between LSW production and changes
in the AMOC. As such, any high-resolution simulation which is capable of resolving the fine
scale features within the Labrador Sea should be carried out for many years to further
understand the climate system. Resolving the full North Atlantic at high resolution (1/60°) and
carrying out a simulation for longer than 10 years would currently be extremely expensive; the
above 1/60° simulations are 5 or so years in length. However, one can incorporate nested
domains to increase horizontal resolution with a relatively minor increase in computing cost.

To simulate the Labrador Sea as accurately as possible, we set up a complex numerical

configuration which achieves very high resolution within the Labrador Sea while keeping
computing costs low such that we will produce over 15 years of simulated data. This simulation
will be kept up to near-present time, lagged a few months depending on the availability of
forcing data. The high resolution allows for explicit representation of eddies which are crucial to
controlling the stratification within the region. We will first describe the model configuration in
detail and then compare against similarly-forced lower-resolution simulations to understand
how changes in horizontal resolution impacts model results in the Labrador Sea.

2.  Methods

The numerical model used for our high-resolution simulation is the Nucleus for

European Modelling of the Ocean (NEMO; Madec, 2008), version 3.6, which is coupled to a sea-
ice model, LIM2 (Fichefet and Maqueda, 1997). The 1/4° Arctic Northern Hemisphere Atlantic
configuration (ANHA4; Fig 1a) is used and includes a double nest via the Adaptive Grid
Refinement in FORTRAN package (AGRIF; Debreu et al., 2008). The AGRIF software allows for
high-resolution nests to communicate along their boundaries, passing information back and
forth between domains. The parent ANHA4 domain extends from Bering Strait, though the
Arctic and North Atlantic, to 20°S in the South Atlantic. The parent domain's nest uses a spatial
and temporal refinement factor of three, bringing resolution to 1/12° and the time step to 240s
(Table 1) in the North Atlantic Sub Polar Gyre domain (SPG12; Fig 1b). An ANHA4 configuration
with a SPG12 nest has been evaluated before by investigating how model resolution influences
Labrador Sea Water formation (Garcia-Quintana et al., 2019) as well as eddy formation and
eddy fluxes in the North Atlantic Current (Müller et al., 2017; Müller et al., 2019). Another nest
is implemented within the SPG12 domain, using a spatial and temporal refinement of five,
increasing the horizontal resolution from 1/12° to 1/60° and reducing the time step to 48s
within the Labrador Sea (LAB60; Fig 1c). All nests allow two-way communication such that the
parent domain supplies boundary conditions while the daughter domain returns interpolated
values to all associated parent grid points. All domains have different horizontal grid spacing
but they share the same vertical grid which is set to 75 geopotential levels (Fig. 1d) using partial
steps (Barnier et al., 2006). This simulation involves three domains (ANHA4, SPG12, and LAB60)
although we primarily discuss what occurs within the 1/60° nest.
A total variance dissipation scheme (Zalesak, 1979) was used in all domains to calculate
horizontal advection. A Laplacian operator was used to compute lateral diffusion in all domains,
while a bi-laplacian operator was used for lateral momentum mixing. As some model
parameters are grid-scale dependent, Table 1 displays these settings. As lateral boundary
conditions have been shown to be very important at producing Irminger Rings in high-
resolution simulations (Rieck et al. 2019), we used no-slip lateral boundary conditions within
the LAB60 domain while the other domains had free-slip conditions. Model mixed layer depths
were calculated via the vertical gradient in temperature and salinity (Holte and Talley, 2009) as
opposed to a 0.01 kg m$^{-3}$ change in potential density between the surface and the bottom of
the mixed layer; the latter method can produce deeper mixed layers than observations suggest
(Courtois et al., 2017). Settings not listed in Table 1 indicate that all domains have an identical
value or option; some of these important settings are shown in Table 2.
Model bathymetry was interpolated from the 1/60° ETOPO GEBCO dataset (Amante and
Eakins, 2009) to each domain's grid and bathymetric smoothing along nest boundaries was
carried out in order to conserve volume where the parent domain supplies boundary conditions
to the daughter domain. All domains were initialized from GLORYS1v1 (Ferry et al., 2009), a
global reanalysis ocean simulation, at the beginning of 2002. Monthly open boundary
conditions (3D T, S, U, V, and 2D SSH and ice values) across Bering Strait and 20° S were
supplied to the ANHA4 domain. These boundary conditions were linearly interpolated from
monthly values, overriding the values within the boundary without the use of a sponge layer.
Runoff was supplied via Dai et al. (2009) while we also included Greenland runoff as estimated
from a surface mass-balance model (Bamber et al., 2012). Without an iceberg model
functioning with the AGRIF software, we treated all solid runoff as a liquid, thus capturing the
full freshwater mass at the cost of accuracy in the spatial and temporal placement of
freshwater emitted from icebergs.

Precipitation, shortwave radiation, downward longwave radiation, 2 meter specific

humidity, 2 meter temperature, 10 meter meridional and 10 meter zonal winds originally were
supplied from the Canadian Meteorological Centre's Global Deterministic Prediction System's
Reforecast product (CGRF; Smith et al., 2014). While high in temporal (hourly) and spatial
resolution (33 km in the Labrador Sea), we found the air-sea fluxes were slightly too weak to
sustain deep convection after 2010. Rather than start completely over, we switched the
atmospheric forcing in 2007 (Fig. 2) when LAB60's mixed layer was still similar to observations.
Starting on 1 Jan 2007, we used the DRAKKAR Forcing Set 5.2 (DFS; Dussin et al., 2016). DFS
supplies data at 3 hour increments for wind, temperature, and humidity, while precipitation
and radiation are daily. DFS has a spatial resolution which is approximately 45 km within the
Labrador Sea. Our own analysis of the CGRF data showed a 2002-2015 average yearly heat loss
of 47 W m$^{-2}$ from the interior Labrador Sea while DFS removed 53 W m$^{-2}$ (Pennelly and Myers,
submitted). Increasing the horizontal resolution likely increased the horizontal buoyancy fluxes
and rendered the CGRF's air-sea heat loss, which was appropriate in our ANHA4 and ANHA12
configurations, inadequate. The decision to swap to DFS was based on its greater heat loss,
promoting a better mixed layer depth throughout the Labrador Sea, though a different forcing
product will eventually be needed as DFS does not currently extend past 2017. Supplemental
Fig. 1 depicts the difference in mixed layer depth between the LAB60 simulation forced by
CGRF, when forced with CGRF through 2007 and then forced by DFS, as well as what ARGO
observations suggest. The weaker air-sea heat loss as forced by the CGRF product leaves the
mixed layer with little interannual variability that doesn't compare well with observations.
Early testing showed that adding passive tracers increases the computing resources
required by about 20% per passive tracer. To keep the simulation from requiring too many
resources, we limited LAB60 to three passive tracers:
1.  Liquid runoff from Greenland
2.  Irminger Water  (T> 3.5°C, S>34.88) which flows westward past Cape Farwell (Fig. 3b)
3.  Labrador Sea Water ($\sigma_\theta$>27.68 kg m$^{-3}$) formed within the mixed layer of the Labrador

Sea (Fig. 3c)

Runoff from Greenland was included due to the importance of Greenland's freshwater
contribution to changes within the Labrador Sea. Water mass definitions for Irminger Water
and Labrador Sea Water were selected based on previous studies (i.e. Kieke et al., 2006; Myers
et al., 2007). Note that there is no maximum density criteria given to our Labrador Sea Water
tracer- the tracer is formed throughout the water column until it reaches the bottom of the
mixed layer. Figure 3 illustrates both the source regions as well as the tracer extent as of 1 Jan
2010. While these water masses have been studied before (Kieke et al., 2006; Myers et al.,
2007; Böning et al., 2016), there has been no attempt to use them as passive tracers at a
resolution higher than 1/20° (Böning et al., 2016).
The LAB60 simulation originally started on the Graham cluster of Compute Canada.
Other high-resolution simulations often use thousands of computer processors but our
simulation could not run on more than 672 CPUs on this cluster as it would stall during domain
construction. The years 2002-2007 were carried out on Graham, after which a new allocation
on a different high performance Compute Canada cluster, Niagara, became available to us. The
LAB60 simulation on Niagara did not suffer from the same issue as it did on Graham and we
were able to use many more processors. Initial testing found a substantial increase in the
number of days simulated per job submission when the number of CPUs was increased from
672 to 3000; tests using 4000 CPUs showed no further improvement. Thus, we carried out the
remainder of the LAB60 simulation with 3000 CPUs. Each job submission required around 22
hours to carry out, providing 40 days of model output. The real time to finish each 40 day
submission naturally varied across the year, increasing during winter which we attribute to the
sea-ice model.
A spin-up period (Fig. 2) was required as the model quickly went unstable and crashed.
We attribute this to the interpolation of the 1/12° GLORYS1v1 data onto the LAB60 grid; the
resulting data were not smooth enough and numerical noise was generated, leading to model
failure. To reduce this noise, a gradual spin-up procedure took place. First, we kept the
numerical timestep very low (2s in LAB60) when the model was initialized. We also set the
1/60° nests' eddy viscosity and diffusivity values to be equal to those within the SPG12 nest.
We gradually increased the timestep and reduced the viscosity and diffusivity values over the
first year (2002) to what is within Table 1. Other than also increasing the timestep to stay in line
with LAB60, no other values were changed across the coarser ANHA4 and SPG12 domains. To
allow LAB60 to adjust to the final settings, we consider the 2003 year to be an adjustment year
(Fig. 2).
To assess the validity of LAB60, model results were compared against AVISO satellite
data (https://www.aviso.altimetry.fr/), specifically U/V geostrophic velocities which are derived
from the sea surface height. Argo profiler data (http://www.argo.net/) was also used to assess
the mixed layer. Bottle data from cruise 18HUD20080520, accessed from CCHDO
(https://cchdo.ucsd.edu/cruise/18HU20080520) on 10 April 2018 was used to compare
observations across the AR7W section.

3.   Model Simulation Results
To understand what is gained by resolving the Labrador Sea at 1/60°, we compare the
output of our LAB60 simulation with similarly forced ANHA simulations at both 1/4° (ANHA4)
and 1/12° (ANHA12). The large-scale circulation (top 50m) is shown for our 3 simulations (Fig. 4)
as well as AVISO geostrophic velocities. All simulations have greater speed within the West
Greenland Current (ANHA4: up to 0.8; ANHA12: 0.8; LAB60: 0.6; AVISO: 0.4 m s$^{-1}$) and Labrador
Current (ANHA4: up to 0.6; ANHA12: 0.6; LAB60: 0.4; AVISO: 0.4 m s$^{-1}$) as altimetry observations
suggest slower speeds here. However, Lin et al., (2018) found maximum speed up to 0.74 m s$^{-1}$
along the west coast of Greenland. Both the ANHA4 and ANHA12 configuration have larger
values further up the western coast of Greenland, as well as connecting the West Greenland
Current and the Labrador Current; features that do not occur in both LAB60 and observations.
As LAB60 and observations have less average speed occurring within these boundary currents,
we suspect that all configurations have some large differences in eddy activity, particularly
where these boundary currents are.

Eddy kinetic energy (EKE: $0.5(\overline{U_g'^2} + \overline{V_g'^2})$, Fig. 5) was calculated from geostrophic

velocity anomaly based on the sea level anomaly (SLA) from the 2004-2013 mean state:

$$U_g' = -\frac{g}{f}\frac{SLA}{\Delta y}$$

$$V_g' = -\frac{g}{f}\frac{SLA}{\Delta x}$$

where g is the gravitational constant, $f$ is the Coriolis parameter, and Δy and Δx are model grid
length. Overbars indicate the 2004-2013 mean value while primed variables indicate a deviation
from the mean state. AVISO observations were already supplied as geostrophic velocities.
High levels of EKE can be found along the west coast of Greenland (Fig. 5), extending into the
interior of the basin around 62° N, as well as along the Labrador coast's shelf break. The path
extending from the west coast of Greenland is mostly due to Irminger Rings which leave this
coast and travel westward (Chanut et al., 2008). While the EKE extending from west Greenland
enters the interior of the Labrador Sea, that which stems from the Labrador coast does not
penetrate far into the interior. The ANHA4 simulation has low EKE along the west coast of
Greenland (around 100 $cm^2 s^{-2}$) and along the Labrador Coast's shelf break (10-30 $cm^2 s^{-2}$). The
ANHA12 simulation shows improvement, having much higher EKE extending from west
Greenland (100-300 $cm^2 s^{-2}$) however the EKE does not quite extend into the interior of the
Labrador Sea but instead remains in the northern Labrador Sea. Furthermore, there is
additional EKE along the Labrador shelf break (30-50 $cm^2 s^{-2}$) compared against ANHA4. The
LAB60 simulation shows further improvement as the EKE signature from the west Greenland
coast is greater (100-1000 $cm^2 s^{-2}$) and now enters into the interior of the Labrador Sea. A
notable increase in EKE also occurs along the Labrador shelf break (100-200 $cm^2 s^{-2}$) and within
the interior Labrador Sea (10-100 $cm^2 s^{-2}$). LAB60 matches well against observations along the
west coast of Greenland and the Labrador shelf break (both above 1000 $cm^2 s^{-2}$) as well as the
interior Labrador Sea (10-100 $cm^2 s^{-2}$). LAB60's higher interior EKE may be partially from
convective eddies that are formed during the wintertime. However, LAB60 has lower EKE within
the Northwest Corner where ANHA4, ANHA12, and the observations exceed 1000 cm$^2$ s$^{-2}$ over a
wide area. LAB60 matches the spatial distribution albeit with reduced EKE.

The differences in the EKE field between these configurations identify that each

simulation is resolving features of varying spatial scales. The ANHA4 simulation, with low EKE
within the Labrador Sea, does not adequately resolve eddies in this region, as illustrated with a
snapshot of normalized model relative vorticity (Fig. 6). However, the larger scale meanders
within the North Atlantic Current are visible. ANHA12 shows a greater degree of mesoscale
features (50 to 500 km), though distinct eddies within the Labrador Sea are also not resolved.
LAB60 resolves eddies along both the west coast of Greenland as well as the Labrador Coast. A
video showing LAB60's normalized relative vorticity is shown in Supplementary Video 1.

A few Irminger Rings are shown in Fig. 7, a snapshot in time from 26 July 2007. A newly

spawned ring (Fig. 7c) shows very strong surface speeds (up 0.6 m s$^{-1}$ for Ring A; Fig. 7a) while
older eddies to the south have reduced speeds (up to 0.3 m s$^{-1}$ for Ring B; Fig. 7a). To
investigate the stratification strength, we calculate the amount of energy needed to produce a
neutrally stratified column extending down to some reference depth, $h$. This proxy, called
convective energy, is given by:

$$Convective\ energy(h) = \frac{g}{Area} \int \int \left[ h\,\rho_\theta(h) - \int_0^h \rho_\theta(z)\,dz \right] dA$$

where $g$ is the gravitational constant, $Area$ is the total surface area over our region of interest
(Fig. 1c), $h$ is the reference depth (2000m used in this study), $\rho_\theta$ $(z)$ and $\rho_\theta$ $(h)$ are the potential
density at each grid cell and the potential density of the grid cell at the reference depth, and $A$
is the surface area of each grid cell. A strongly stratified column of water corresponds to a high
convective energy value. A snapshot of convective energy (Fig. 7b) shows that most of these
eddies have substantially higher amounts compared to the background Labrador Sea,
suggesting that the cool and fresh WGC water, as well as warm and salty Irminger Water keep
these eddies strongly stratified. However, these eddies age within the Labrador Sea, and while
a new eddy has strong stratification (>3000 J m$^{-3}$), an eddy which has evolved over many
months (Fig. 7d) has weaker stratification (about 2000 J m$^{-3}$). Older eddies may have very weak
stratification as they may have experienced two convective winter periods of buoyancy
removal. This has been noted before, as Lilly et al. (2003) found aged Irminger Rings with a
mixed layer that surpassed 1000m.

These differences in resolving the mesoscale (50 to 500 km) and sub-mesoscale (<50

km) processes within each simulation produced significant changes within the Labrador Sea as
seen from modeled convective energy values as averaged from 2004-2013 (Fig. 8). Resolving
few eddies, the ANHA4 simulation's interior Labrador Sea lacks the buoyancy flux and remains
very weakly stratified across a wide region. The ANHA12 simulation partially resolves some
mesoscale features and eddy fluxes from the Greenland coast which supplies buoyancy to the
Northern Labrador Sea and has higher convective energy. Furthermore, the spatial extent of
the weakly stratified region has shrunk and resides primarily within the Labrador Sea, as
opposed to ANHA4 which spills out of the basin. LAB60, fully capable of resolving buoyant
eddies from the Greenland and Labrador coast, as well as convective eddies, has a much
stronger degree of stratification in the interior region. A visible path of strong stratification
appears around 60°N along this coastline, eventually extending away from the coastline around
62°N. This path is consistent with the general path that simulated Irminger Rings take (Chanut
et al., 2008). Supplemental Video 2 shows the convective energy of the LAB60 simulation from
2004 through the end of 2013.

The ANHA4 simulation experiences weaker stratification in the Labrador Sea than

ANHA12 and LAB60, driving a deeper maximum mixed layer that also covers a larger spatial
extent (Fig. 9). However, the maximum mixed layer depth as simulated by ANHA4 and ANHA12
greatly exceed what Argo observations suggest (Fig. 9d). ANHA12 has higher EKE within the
WGC, supplying more buoyancy to the northern portion of the Labrador Sea, reducing both the
vertical extent of the mixed layer as well as the spatial extent where the mixed layer is deeper
than 1000m. LAB60 has higher EKE than ANHA12, and the vertical and spatial extent of deep
mixing is reduced even further. LAB60's mixed layer is far more similar to what ARGO
observations suggest, suggesting the additional eddy fluxes to be fairly accurate. The evolution
of LAB60's mixed layer depth is shown in supplemental video 3 from 2004 through the end of

2013.

After the bottom of the mixed layer returns to the near-surface, a newly formed LSW
mass is left behind. To account for density drift, we allow the LSW classification to evolve in
time, unlike our LSW passive tracer. We calculated LSW density and thickness by binning by
potential density, referenced to 1000 dbar, with bin lengths of 0.001 kg m$^{-3}$. This was carried
out within the black outlined polygon in Fig 1c for each daily output file per year. The density
bin which had the thickest layer across the year was set as the maximum density of LSW for
that year. The minimum density was defined to be 0.02 kg m$^{-3}$ less than the maximum density.
Linear interpolation occurred between years to allow for a gradual shift in density to prevent
staircase patterns from emerging. Large differences in both the density as well as the thickness
are present between the simulations shown in Fig. 10. The ANHA4 and ANHA12 simulations
have similar density values of LSW while the LAB60 simulation is less dense. While the
interannual variability matches fairly well across all configurations, the density values suggested
by LAB60 are closer to ARGO observations (32.34 to 32.36 kg m$^{-3}$; Yashayaev and Loder, 2016)
during the same time period. We suspect the denser LSW formed by ANHA4 and ANHA12 is
primarily attributed to the lack of buoyancy coming from Greenland. As similar air-sea heat
losses should occur in all three configurations, the weaker stratification of ANHA4 and ANHA12
indicates that deep mixing is more likely producing not only a denser LSW layer, but also a
thicker one. Yashayaev and Loder (2016) also investigated the thickness of LSW (their Fig. 8),
and while our simulations do not quite capture the same interannual variability and amplitude
suggested their analysis using ARGO profilers, LAB60 is far more accurate than the lower-
resolution configurations.
All simulations encounter some degree of numerical drift within the Labrador Sea (Fig.
11), judging from the salt and heat content change as calculated between the surface and
seafloor within the polygon in Figure 1 since 2004. ANHA4 experiences the largest drift in both
salt and heat, helping us understand why LSW is so dense in this simulation. ANHA12 also
experiences drift, though slightly less severe. LAB60 has a small but gradual increase in both salt
and heat content although it is difficult to state if this is drift or simply interannual to decadal
variability. Regardless of the cause, LAB60's change in both heat and salt content is very
minimal compared against the lower-resolution simulations.
When compared against bottle data collected during a single hydrographic cruise across
Atlantic Repeat Hydrography Line 7 West (AR7W; Fig 12), LAB60 is slightly warmer (about 0.25
°C) and saltier (about 0.05 kg m$^{-3}$) throughout the interior. This causes LAB60 to be slightly
denser with isopycnals residing higher than observations during this cruise suggest.
Observations were not carried out above Greenland's continental slope, although they show
some presence of the warm core of the WGC which the model captures.  Salinity values close to
the Labrador coast compare well while LAB60 is slightly warmer (about 0.5 °C) above the
continental shelf.
The three passive tracers implemented within the full LAB60 configuration (Fig. 3) show
where Greenland runoff, Irminger Water, and Labrador Sea Water travel to. These tracers were
selected because they either contain a significant amount of buoyant water compared to the
Labrador Sea, or are produced via convection in the Labrador Sea. From this image on 1 Jan
2010, we see a large portion of Greenland's runoff (Fig. 3a) resides within Baffin Bay as well as
along the Labrador Coast. Some of this tracer is present where the ocean depth is greater than
2000m. A few Irminger Rings are identifiable, due to their thicker freshwater cap, which are in
water deeper than 3000m. Little exchange with the interior basin appears to occur along the
Labrador Current until the vicinity of Flemish Cap, after which a significant portion of the tracer
propagates eastward. Supplemental Video 4 shows this the evolution of this tracer from 2004
through the end of 2013.
Irminger Water (T>3.5°C, S> 34.88; Fig. 3b) which flows west past Cape Farwell, enters
the interior Labrador Sea with the greatest amounts where the seafloor is at a depth between
2000 and 3000m. Similar as above, individual Irminger Rings are visible, containing a larger
amount of Irminger Water than the surrounding water. This water mass also flows along the
Labrador Coast until it is in the vicinity of Flemish Cap. Supplemental Video 5 shows this the
evolution of this tracer from 2004 through the end of 2013.
Our Labrador Sea Water tracer (Fig. 3c) is traced where the mixed layer produces water
with a potential density above 1027.68 kg m$^{-3}$ within the black contour identified in the figure.
This definition differs compared to our method of classifying LSW as we did not implement any
FORTRAN code to detect and compensate for density drift of our simulation, instead sticking to
a strict density classification for this tracer. As this image was made at the start of the
convection season, the current deep patch is a freshly produced layer that reaches up to 800m
deep. After forming, LSW spreads southwards along the Labrador shelf break as well as to the
southeast. Supplemental Video 6 shows this the evolution of this tracer from 2004 through the
end of 2013.
4.   Discussion
We describe a 10+ year long, high-resolution simulation which achieves 1/60° horizontal
resolution in the Labrador Sea via two nests inside a regional configuration, resolving mesoscale
and sub-mesoscale processes which strongly impact the deep convection which occurs here.
We show that lower-resolution simulations fail to resolve these key processes that strongly
control the production of Labrador Sea Water, an important water mass within the Atlantic
Meridional Overturning Circulation. While the NATL60 and eNATL60 simulations were designed
with the SWOT altimetry satellite mission in mind (NATL60 website: https://meom-
group.github.io/swot-natl60/virtual-ocean.html), their integration period, like many other high-
resolution simulations, is a handful of years. LAB60, although covering a much smaller region,
could be a valuable asset to many users who require a lengthy period of high-resolution model
output. We also have included three passive tracers which are often excluded in simulations at
this resolution. Our three passive tracers highlight regions where each water mass enters the
interior region of the Labrador Sea, demonstrating the pathways of buoyant Greenland melt
and Irminger water. Furthermore, we trace Labrador Sea Water which is formed during the
convective winter period.
We show that LAB60 has greater EKE than our lower-resolution simulation, resolving
eddy fluxes including Irminger Rings, boundary current eddies, and likely convective eddies as
indicated by greater EKE within the interior. Boundary current eddies still appear relatively
disconnected from the interior basin, adding further support that these eddies have limited
influence on convection and restratification (Rieck et al., 2019). We offer no additional support
regarding the relative importance of Irminger Rings and convective eddies on controlling deep
convection; this is currently being investigated for a later manuscript. Model drift appears very
low, a large improvement over the ANHA4 and ANHA12 configurations. The drift might produce
slightly denser LSW than observations suggest, however LAB60s density is much more accurate
than ANHA4 and ANHA12. The boundaries of LAB60, supplied by the inner SPG12 nest, may
influence the high-resolution nest. We note that the North Atlantic Current, which is close to
the boundary, has less EKE and vorticity than the ANHA4 and ANHA12 simulations. Conversely,
the WGC close to the eastern nested boundary has multiple jets which have been noted in
hydrographic data (Pickart, personal communication). Boundary communication is always a
concern in nested simulations and LAB60 is no different. More investigation will reveal any
potential boundary issues but our results so far indicate no further areas of potential concern.

Others have investigated the Labrador Sea using numerical simulations with different

resolution. Böning et al. (2016) traced Greenland meltwater with the 1/20° VIKING20 and 1/4°
ORCA025 simulations, noting more meltwater entered the interior Labrador Sea at higher
resolution partially as a result of greater WGC eddy fluxes but not from the Labrador coast. The
minor amount of eddy fluxes from the Labrador coast has been noted earlier even at lower
resolution (1/3°; Myers, 2005). Steadily increasing horizontal resolution has so far not changed
this for the Labrador coast, though this is opposite for the WGC. LAB60 has a clear increase in
EKE and likely greater eddy fluxes from the WGC into the interior of the Labrador Sea.

We have many ambitious research topics which we plan to use LAB60 to investigate.

This includes, but is not limited to, the variability and structure of the West Greenland Coastal
Current, Labrador Sea Water production, and the role of both Irminger Rings and convective
eddies in controlling stratification in the Labrador Sea. This lengthy high-resolution simulation
with three passive tracers will provide valuable information for many numerical studies within
the Labrador Sea for years to come.

Code and/or data availability

The FORTRAN code used to carry out the LAB60 simulation can be accessed from the

NEMO version 3.6 repository
(https://forge.ipsl.jussieu.fr/nemo/browser/NEMO/releases/release-3.6). A few FORTRAN files
were modified to handle our passive tracers. The complete FORTRAN files as well as the
CPP.keys, namelists, and associated files can be found on Zenodo (Pennelly, 2020). Initial and
boundary conditions, atmospheric forcing, and numerical output were too large to host on a
repository and instead are hosted on our lab's servers as well as the Compute Canada Niagara
server. These data can be requested by emailing the corresponding author.

Author Contribution
PM designed the layout of the LAB60 configuration which included the region of
interest, numerical length, and which forcing and initial conditions to supply, as well as
supervised CP. CP produced the configuration, modified the FORTRAN code, set up the
configuration on the high-performance computing systems, carried out the simulation, and
performed the analysis. The manuscript was prepared by CP with contributions by PM.

Acknowledgements
The authors would like to thank the NEMO development team as well as the DRAKKAR
group for providing the model code and continuous guidance. We express our thanks to
Westgrid and Compute Canada (http://www.computecanada.ca) for the computational
resources to carry out our numerical simulations as well as archival of the experiments. We
would like to thank Nathan Grivault for his help to migrate our configuration between
computing clusters, as well as Charlene Feucher for her help with ARGO data. This work was
supported by an NSERC Climate Change and Atmospheric Research Grant (Grant RGPCC
433898) as well as an NSERC Discovery Grant (Grant RGPIN 04357).

The authors declare that they have no conflict of interest.

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

Tables

Table 1: Domain settings for the ANHA4 parent domain, SPG12 and LAB60 nested domains.
Other settings which are invariant to the domain are shown in Table 2.

| Setting | ANHA4 | SPG12 | LAB60 |
|---|---|---|---|
| Horz. Resolution | 1/4° | 1/12° | 1/60° |
| X points | 544 | 724 | 1179 |
| Y points | 800 | 694 | 2659 |
| Timestep [ s ] | 720 | 240 | 48 |
| Horiz. Eddy Viscosity [ $m^4 s^{-1}$ ] | 1.5x10^11 | 1.5x10^10 | 3.5x10^8 |
| Horiz. Eddy Diffusivity [ $m^2 s^{-1}$ ] | 300 | 50 | 20 |
| Lateral Slip Conditions | Free slip | Free slip | No slip |














Table 2: Model configuration settings which are identical between all three domains. **Bold**
values indicate values which were changed when we migrated LAB60 from the Graham cluster
to Niagara.

| Configuration Setting | Value |
| --- | --- |
| Vertical grid | 75 geopotential levels |
| Sea-ice model | LIM 2 (Fichefet and Maqueda, 1997) |
| Bulk formula | CORE (Large and Yeager, 2008) |
| Liquid discharge | Dia et al. (2009) + Bamber (2012: Greenland) |
| Solid discharge | Input as liquid |
| Surface Restoring | None |
| Initial conditions | Glorys1v1 (T,S,U,V,SSH,ice) |
| Open boundary conditions | Glorys1v1 (T,S,U,V,ice) |
| Atmospheric forcing: | |
| 2002-2006 | CGRF (Smith et al, 2014) |
| 2007-2017 | Drakkar Forcing Set 5.2 (Dussin et al. 2016) |
| Lateral momentum | Bilaplacian operator |
| Lateral diffusion | Laplacian operator |
| Vertical eddy viscosity | $1\times10\text{^}-4 \, m^2 \, s^{-1}$ |
| Vertical eddy diffusivity | $1\times10\text{^}-5 \, m^2 \, s^{-1}$ |
| Mixed layer scheme | Holte and Talley (2009) |
| Bottom friction | Nonlinear |
| Hydrostatic approximation | Yes |
| Passive tracers | Three (see Figure 2) |
| CPU requested | 672 (**3000**), Broadwell 2.1 GHz (Skylake 2.4 GHz) |
| Time to complete 1 year | Approximately 700 (**200**) hours |
| Initialization date | January 1st, 2002 |



Figures

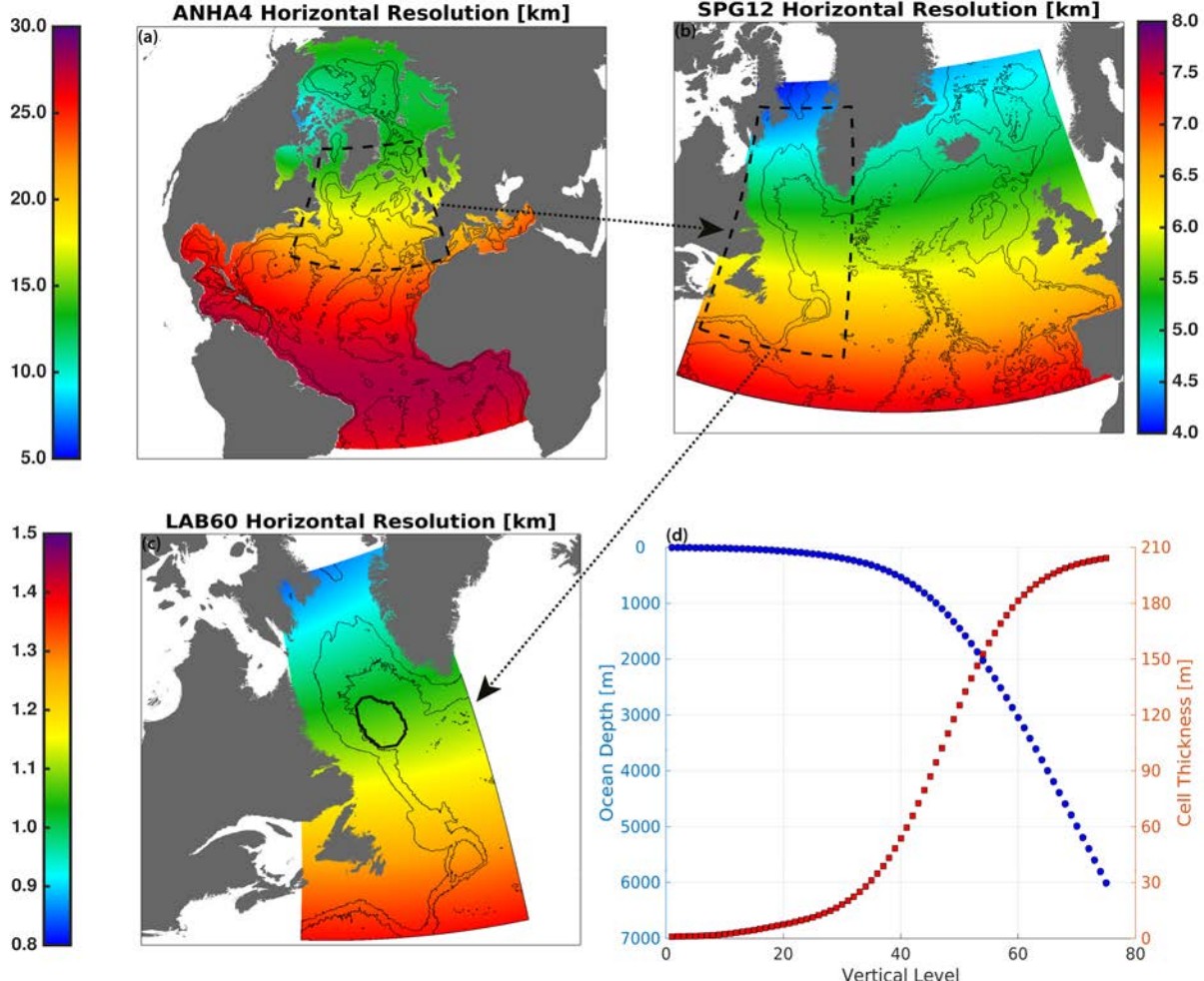

Figure 1: Domain setup for the (a) ANHA4 parent domain, (b) the SPG12 nest, and (c) the LAB60 nest. Horizontal grid resolution, in km, is identified by color. All domains share identical vertical grid structure (d). The thick black contour in (c) identifies a region of interest where calculations of LSW's density, thickness, and mixed layer depth are determined. The 1000m, 3000m, and 5000m isobaths are shown via the thin black contours.

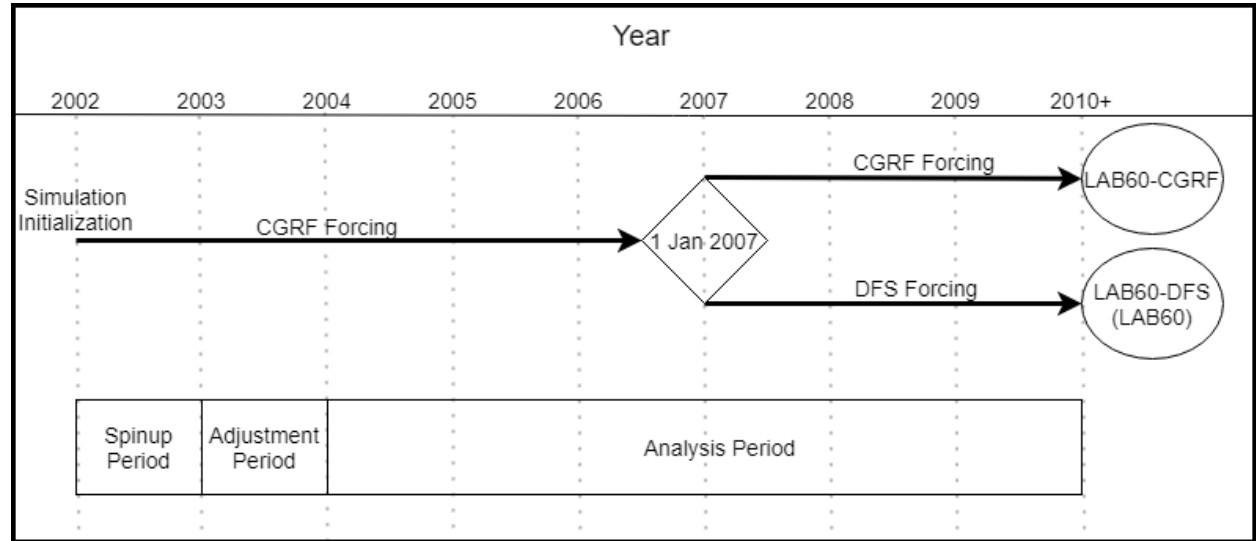


Figure 2: Diagram showing the multiple periods of the LAB60 simulation. The original simulation
was initialized with CGRF atmospheric forcing in 2002, although a branch swapping to DFS
occurred at the start of 2007. This DFS branch is what is primarily presented in this study.

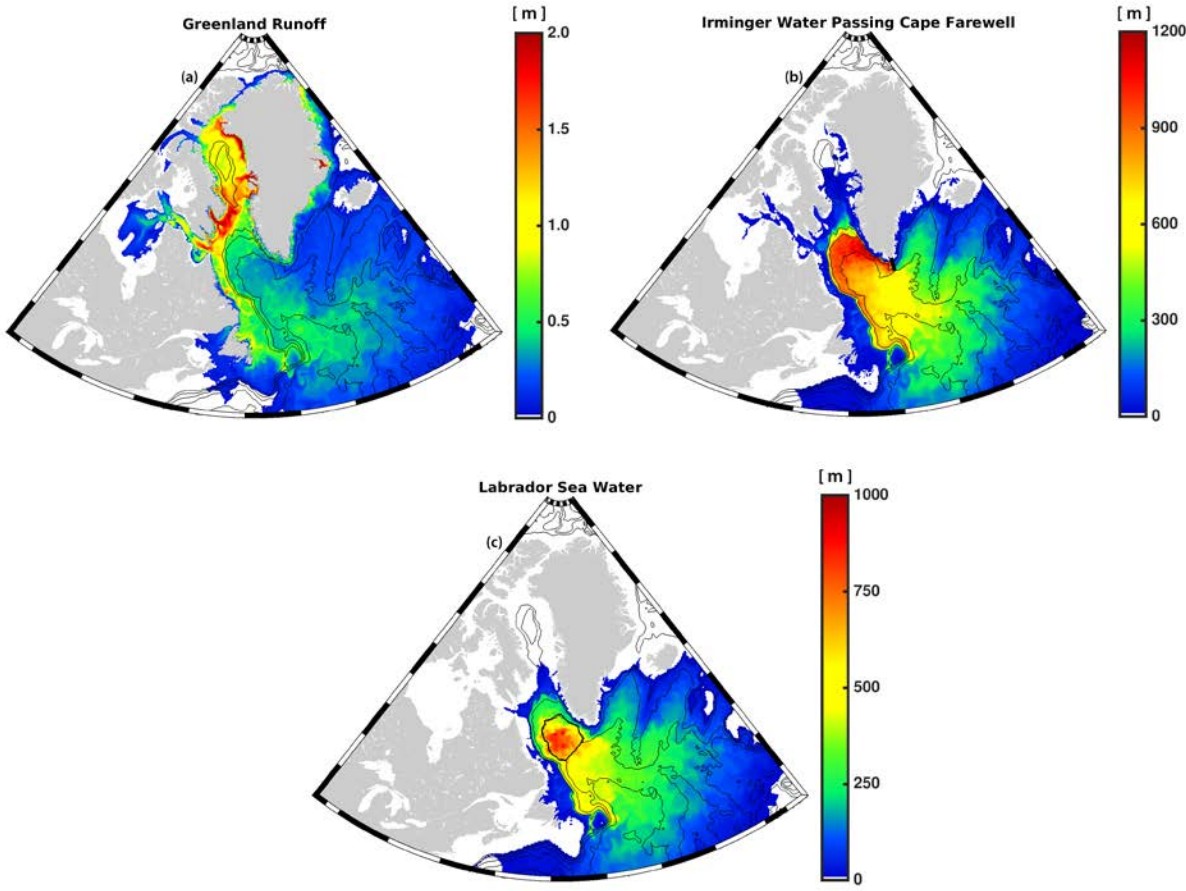


Figure 3: The three passive tracers used within our LAB60 simulation with source regions
indicated by thick black lines: (a) Greenland runoff, (b) Irminger Water (T > 3.5°C, S > 34.88)
which flows west past Cape Farwell, and (c) Labrador Sea Water ($\sigma_\theta$>27.68 kg m$^{-3}$) produced
each convective season. Images are from the simulation date 1 Jan 2010.  Bathymetric contours
are every 1000m. Units are the thickness, in meters, of the tracer. Note: as all three domains
are included in this figure, spatial resolution changes within each subfigure.

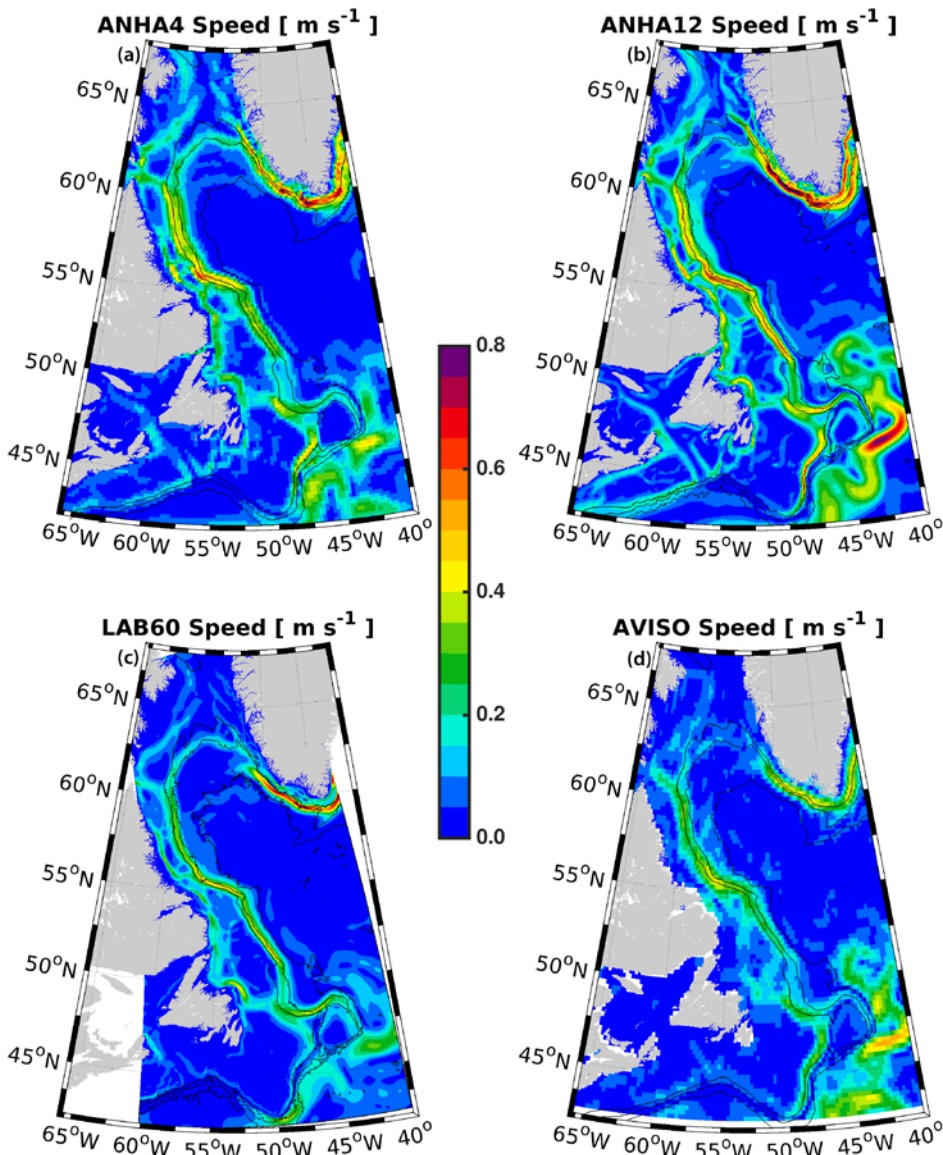


Figure 4: Top 50m average speed (2004-2013) for the (a) ANHA4, (b) ANHA12, (c) and LAB60
simulations, as well as (d) from AVISO observations. The 1000, 2000, and 3000m isobaths are
shown by the black contour lines.





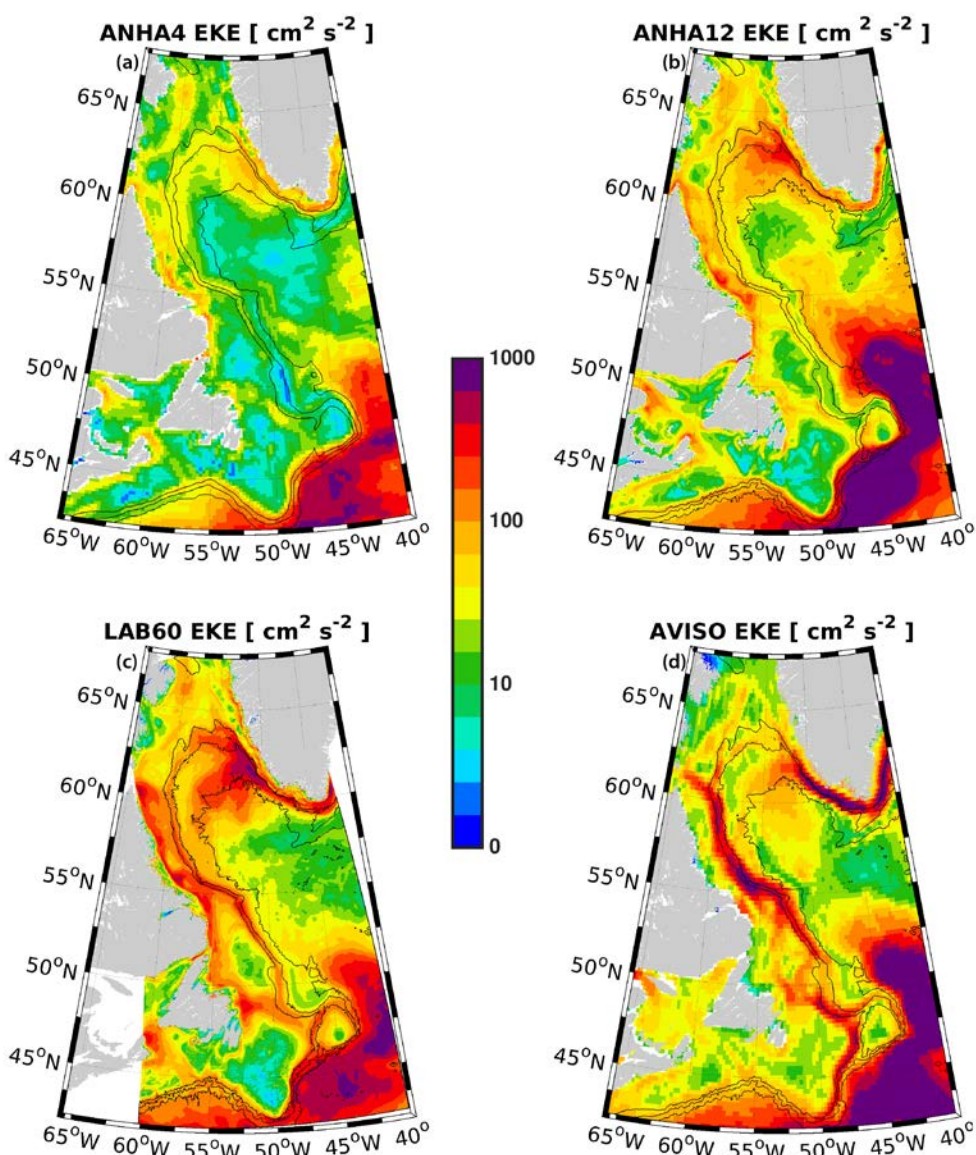


Figure 5: Eddy kinetic energy (EKE), as calculated from geostrophic velocities resulting from the
sea level height anomaly, are shown for (a) ANHA4, (b) ANHA12, and (c) our LAB60 simulation,
from 2004 to 2013. Observations via AVISO are identified in (d). The 1000m, 2000m, and 3000m
isobaths are shown by the black contour lines. A log scale was used for clarity.


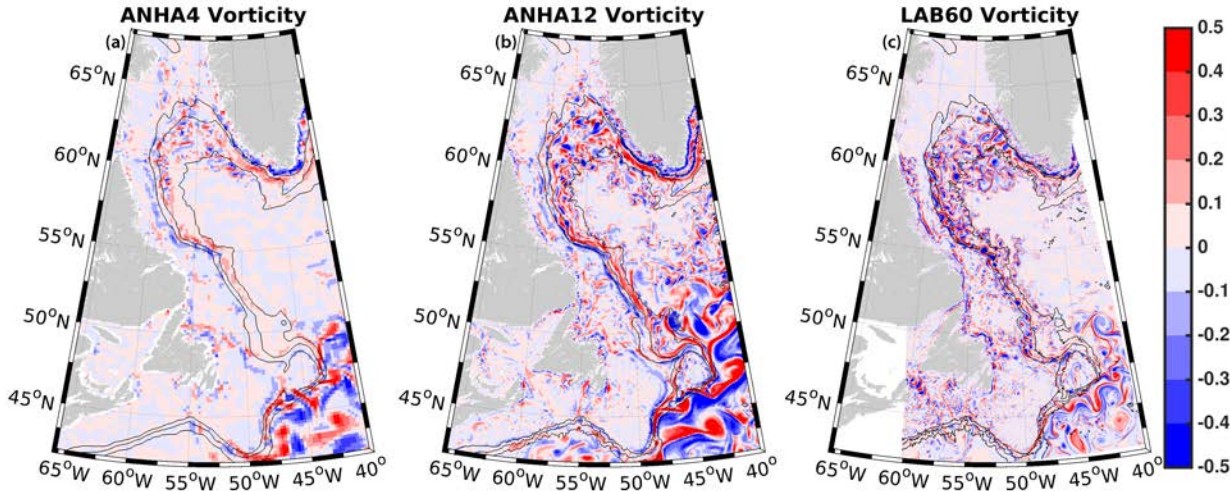


Figure 6: Top 50m relative vorticity, normalized by the planetary vorticity, as simulated by (a)
ANHA4, (b) ANHA12, and (c) LAB60 on 16 March 2008. The 1000m, 2000m, and 3000m isobaths
are shown by the black contour lines.

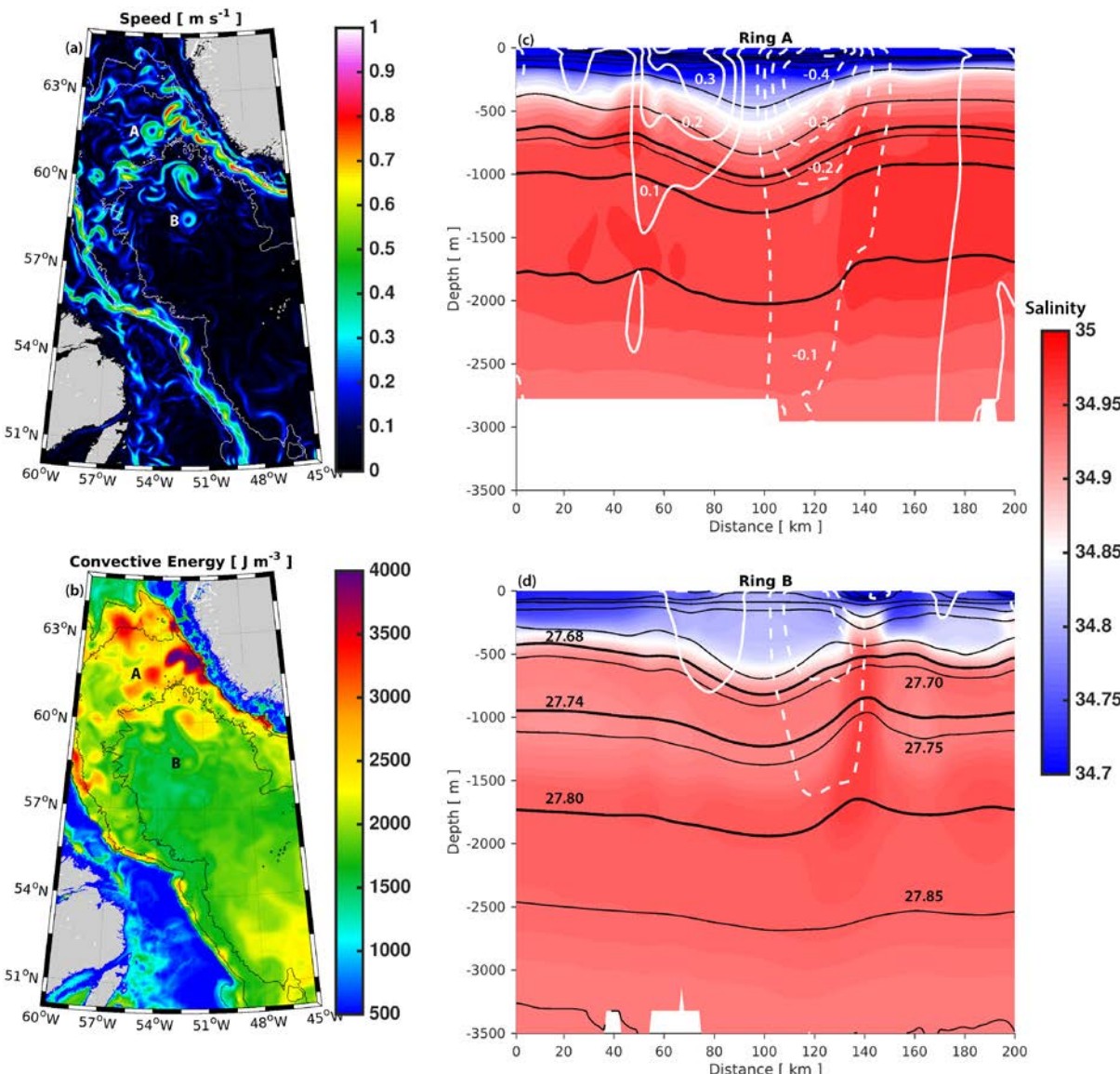


Figure 7: LAB60 snapshot (26 July 2007) of the surface speed (a) and convective energy (b)
within the Labrador Sea. Two Irminger Rings are identified by their age with letters: Ring A is a
young Irminger Ring, while Ring B is comparatively older. An east-west cross section through
each of these Irminger Rings is shown in (c) and (d) where colors indicate salinity, black
contours indicate potential density using a contour interval of 0.05 kg m$^{-3}$, and white contours
indicate meridional velocity where southern flow is dashed and northern flow is solid, using a
contour interval of 0.1 m s$^{-1}$. Thick black contours indicate the potential density classification of
Upper Labrador Sea Water ($\sigma_\theta$=27.68 to 27.74 kg m$^{-3}$) and Classical Labrador Sea Water ($\sigma_\theta$=
27.74 to 27.80 kg m$^{-3}$).

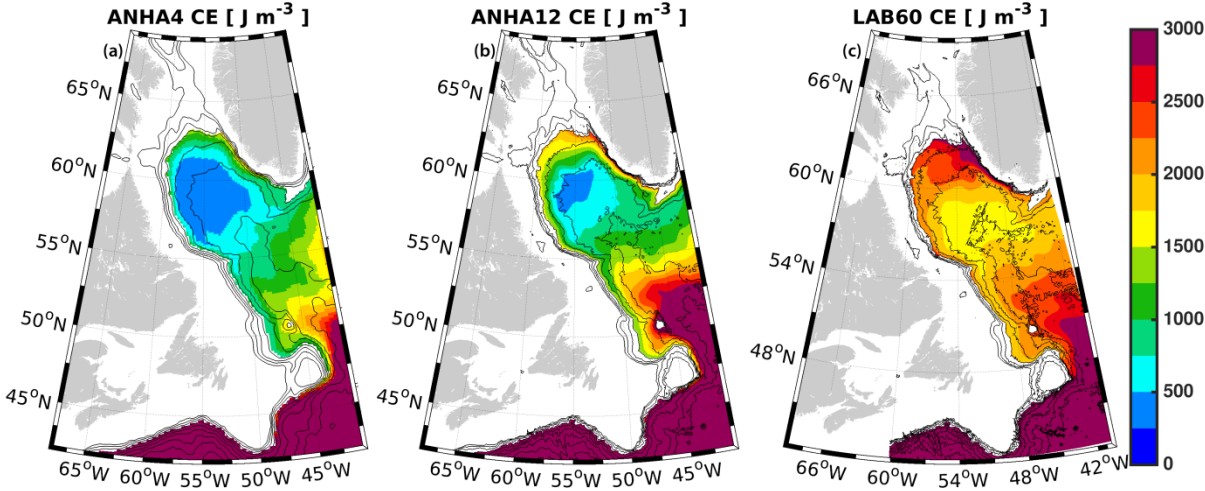


Figure 8: Convective energy (CE), the strength of stratification down to a reference depth of
2000m, is shown for (a) ANHA4, (b) ANHA12, and (c) LAB60. Convective energy was averaged
from 2004 through 2013. Values where the depth of the seafloor was less than 2000m were
removed to preserve clarity. Bathymetric contours (black lines) are shown every 500m.

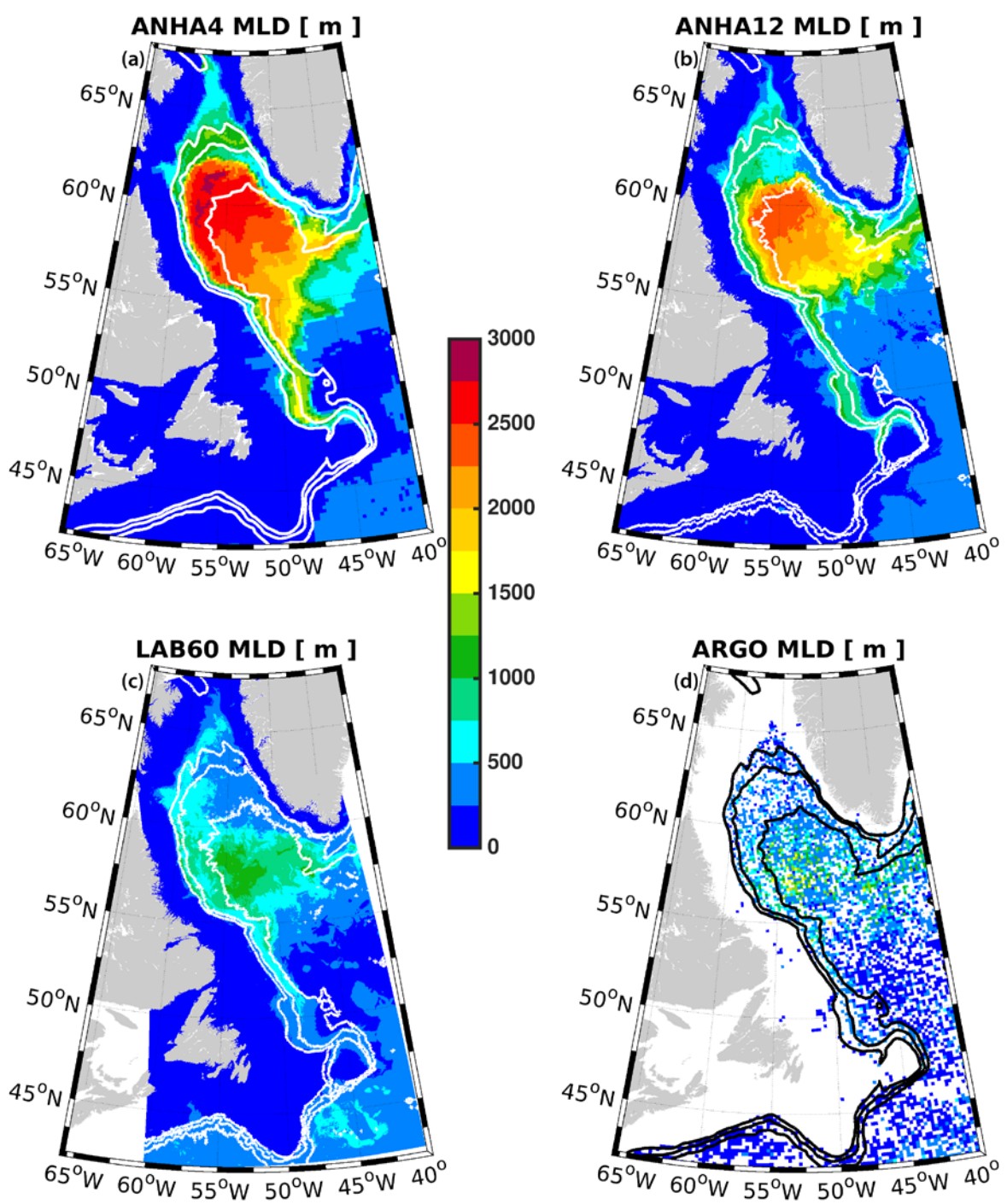

Figure 9: Maximum mixed layer depth for (a) ANHA4, (b) ANHA12, (c) LAB60, as well as (d)
ARGO observations, where available, from 2004 through the end of 2013. For clarity, the ARGO
data were placed on the same grid as ANHA4. The 1000m, 2000m, and 3000m isobaths are
shown via the white and black contours


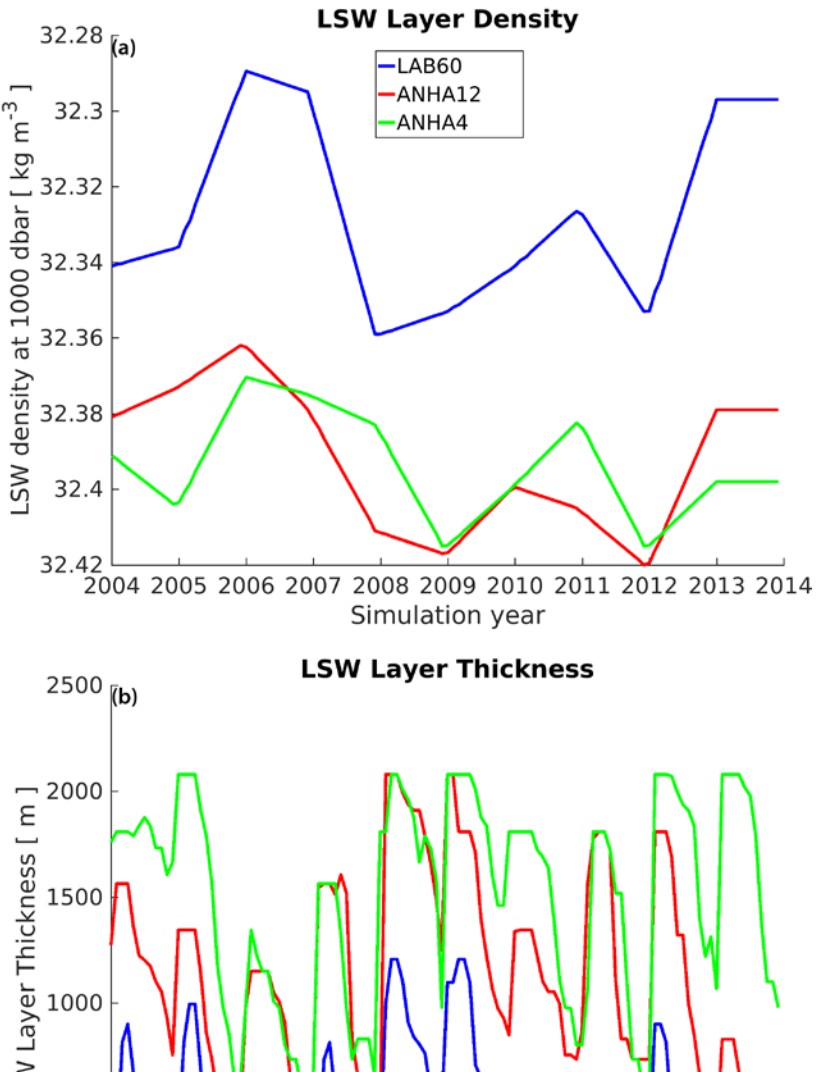


Figure 10: Labrador Sea Water (LSW) density (a) and thickness (b) for the LAB60, ANHA12, and
ANHA4 configurations. LSW density was determined from the thickest layer where a 0.001 kg
m$^{-3}$ change in potential density (ref: 1000 dbar) occurred within the black polygon outlines in
Fig 1c. The LSW layer was then calculated between this density and one which was 0.02 kg m$^{-3}$
less dense.

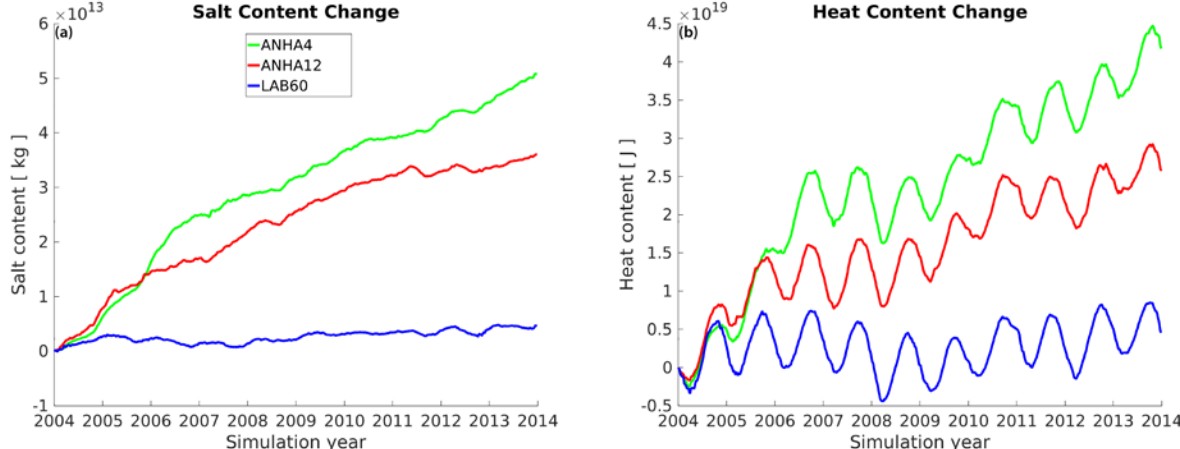


Figure 11: Numerical salt (a) and heat (b) drift in our three simulations as they evolve since 1
Jan 2004. Salt and heat content is calculated over the full ocean column within the polygon in
Fig. 1c.

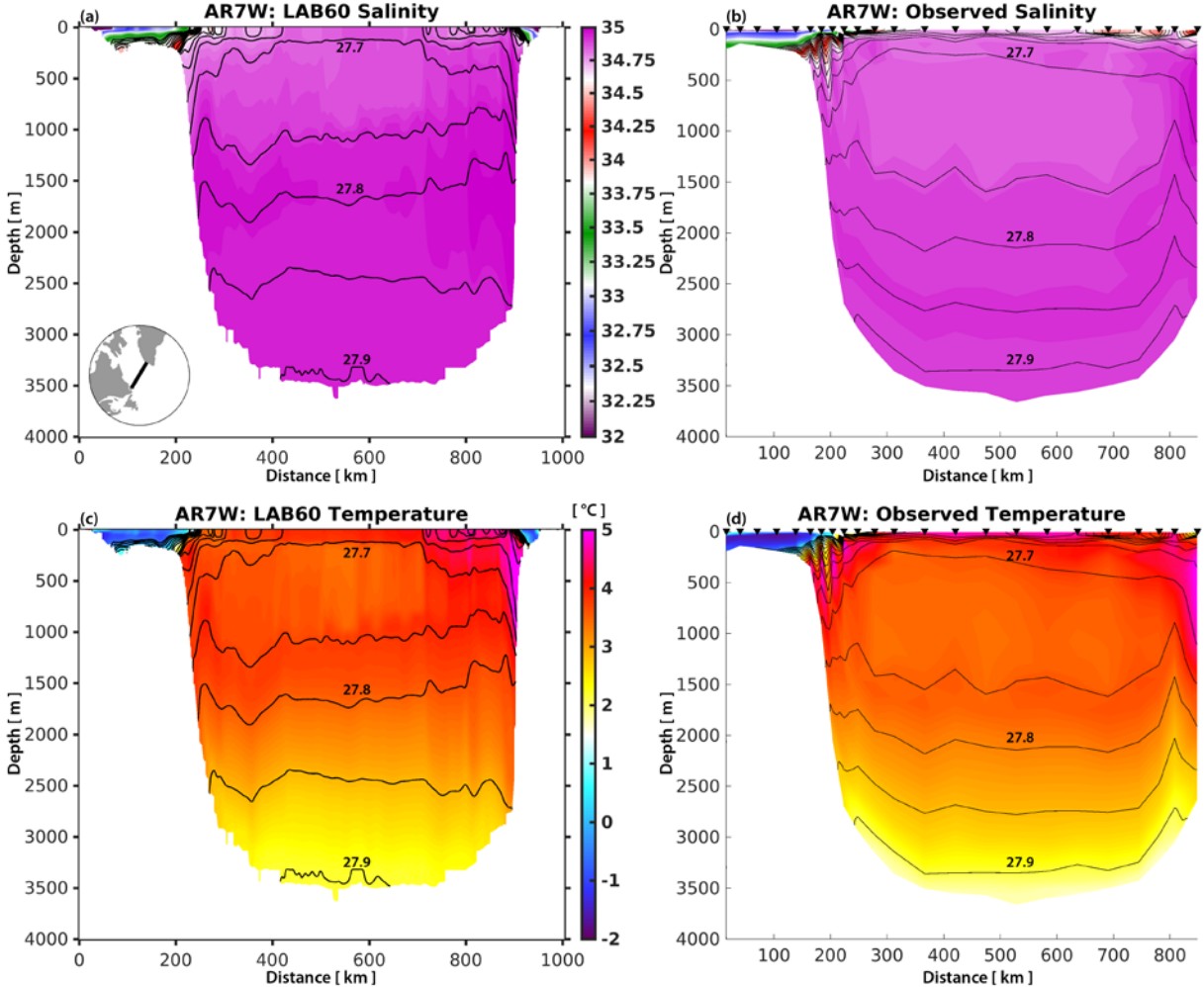


Figure 12: Salinity (top) and temperature (bottom) section across AR7W as determined by the
LAB60 simulation (left) and observations (right) from May 2008. Downward triangles identify
collection sites across the AR7W transit carried out by the CCGS Hudson. Potential density
(black contours) isopycnal interval is 0.05 kg m$^{-3}$.