# Peer review of "Introducing LAB60: A 1/60° NEMO 3.6 numerical simulation of the Labrador Sea Clark Pennelly1\* and Paul G. Myers1"

_Geoscientific Model Development, 2020_

## Referee Comment (RC1) · Anonymous Referee #1 · 8 Jun 2020

**1   General comments**

This study presents first results of a 10-year long nested high-resolution ($\sim$1 km) ocean hindcast run focused on deep convection in the Labrador Sea and the associated mixed layer depth (MLD) restratification and water mass composition. The authors find that small-scale eddies transport buoyant water masses of different origin into the Labrador Sea interior which increase stratification. A marginally coarser horizontal resolution ($\sim$5 km) leads to a significant reduction of these effects and yield too dense waters and too deep MLDs compared to observations.

[Figure]

This model experiment description paper highlights the importance of the ocean model resolution to adequately represent the interplay of small-scale dynamics and the large-scale water mass composition and is hence of interest for the climate community. I consider the overall quality good and suggest this paper accepted with minor revisions.

**2 Specific comments**

62-66: In this sentence, I find it difficult that you 1) use model results only to explain a real world phenomenon and 2) cite yourself only.

73-77: There are other model studies showing the opposite, see e.g. Cael and Jansen (2020) and references therein.

130: It is not discussed at all why a factor of 5 is used to obtain the 1/60° horizontal resolution. Please add this.

Fig. 3 and 4: The LAB60 North Atlantic Current seems to be less vivid and eddy-rich compared to both ANHA12 and AVISO. Can you discuss this?

The "Discussion" section is rather a summary than a discussion. A discussion section is not explicitly required (https://www.geoscientific-model-development.net/for_authors/manuscript_preparation.html → Manuscript composition), so either rename the section or add a discussion. Personally, I would like to see a discussion. For instance, the videos indicate that the LAB60 setup exhibits a model drift and is far from equilibrium (drag the slider of the video player with the mouse from start to end rather fast and you can see a large-scale accumulation of the runoff and Irminger Water tracers in the respective videos). Would a potential model drift influence the LSW time series or the described eddy dynamics? In addition, the shown model data is particularly suited to discuss ongoing questions about meso- and submesoscale energy transfers during convective/unstable situations. Furthermore, your results indicate that

even if an ocean model performs under a high spatial resolution, buoyant water needs to be provided by e.g. the boundary current in the first place to be available for eddies. If this is the case, a nested model configuration like the presented one would have a severe handicap given the coarser resolutions of the parents which provide the boundary current.

**3 Technical corrections**

- sections are not numbered

- its de Steur et al. 2009 (not 2018), Treguier (not Trequier) and Yashayaev (not Yashauaev); Fi* references not in alphabetical order

- 125 & 130: Please specify "temporal refinement".

- Table 2: Please reference LIM2 and CORE as you did for other settings.

- Table 2: Please add the atmospheric forcings (CGRF and DFS) and their respective time periods

- 160-185: LAB60 was forced by CGRF from 2002-2006 (5 years) and by DFS from 2007-2011 (5 years). If this is correct, I find it difficult to plot one LAB60 time series in Fig. 9. Can you at least indicate the two different forcing data sets in the plot/caption?

- 189 & 329: Please specify which sigma/density is used.

- 202: Did the simulation length really increase or rather decrease when the number of CPUs increased?

- 225: configuration"s"?

- 232: Please define how you computed the eddy components (e.g. $u'$) in the model and AVISO data.

- Fig. 5 and Video 1: It is more informative to show relative vorticity normalized by the planetary vorticity, $\zeta/f$, to learn about the transition from meso- to sub-mesoscales.

- 250 & 265: Please define meso- and sub-mesoscales in the introduction and how you separate them.

- 256: Please define convective energy (CE) including the mixing depth to which you refer to in Fig. 6 b. On this snapshot, all CE values are >= 500 J m$^{-3}$. However, a winter situation is shown (17 January 2003) and I would expect CE values < 0 J m$^{-3}$ indicating unstable situations. Is this a misunderstanding?

- 295: First, you find the LSW with $\sigma_\Theta$ > 27.68 kg m$^{-3}$ (line 189; I assume you used $\sigma_\Theta$ in this step). Then, you calculate $\sigma_1$ of this water mass. Then, I don't understand why you define the "yearly maximum density of this water mass as the thickest depth where the density changes by 0.001 kg m$^{-3}$". I don't understand how a "depth" can be "thick" and to which density the 0.001 kg m$^{-3}$ change refers to. Can you reformulate this here and in the caption of Fig. 9? Formulated as it is, I would have problems reproducing this quantity.

- Videos: Adding the sea ice edge and some MLD contours would make the videos even more helpful.

**4 References**

Cael, B. B, and M. F. Jansen (2020): On freshwater fluxes and the Atlantic meridional overturning circulation. Limnology and Oceanography Letters 5, 2020, 185–192, https://aslopubs.onlinelibrary.wiley.com/doi/full/10.1002/lol2.10125

---

## Referee Comment (RC2) · Jan Klaus Rieck (Referee) · 8 Jun 2020

**General comments**

This study presents LAB60, a model experiment based on a new configuration developed with the NEMO ocean model (version 3.6), coupled to the sea ice model LIM2. The configuration consists of a domain covering the Labrador Sea at 1/60° horizontal resolution, nested into a domain covering the subpolar gyre of the North Atlantic at 1/12°, nested into a regional model of the Arctic and North Atlantic at 1/4°.
As noted by the authors, there are other models at such high resolutions of the North Atlantic (NATL60 [Fresnay et al., 2018] and eNATL60 [Le Sommer et al., in prep.], both

$1/60°$ based on NEMO, and a $1/50°$ HYCOM simulation by Chassignet and Xu [2017]). Also the concept of a nested domain within a nested domain in NEMO (using AGRIF) to achieve $1/60°$ resolution is not novel (see for example Schubert et al., 2019). However, the domain of very high resolution is restricted to the Labrador Sea and thus three passive tracers can be implemented and the simulation is planned to be run for more than ten years. These novel features of a simulation at $1/60°$ clearly make this manuscript a valuable contribution to foster the understanding of the Labrador Sea, its variability and the mesoscale's (and possibly sub-mesoscale's) impact on water mass formation and transport.

The manuscript is well structured and comprehensive in most parts, however I deem some major improvements by the authors necessary before publication:

1. You emphasize the need for a long simulation to study the decadal variability of the Labrador Sea and state that your simulation is suitable for that. However, at least at the time of submission, there were only 7 years (excluding spin-up) of the simulation finished and your statement of having a simulation of more than ten years do not hold. If the simulation does now (time of revision) extend over the stated period, this is fine, if not there need to be adjustments to the manuscript. Additionally, you should think about rephrasing the manuscript at some points concerning the suitability of a ~10 year-long simulation to study decadal variability.

2. In my eyes, you do not satisfactorily advertise the advantages and improvements of LAB60 in comparison to other, existing high-resolution simulations carried out with the same model (e.g. the VIKING20X simulations at $1/20°$ [Rieck et al., 2019]). I do think that the presented LAB60 simulation has advantages over the other simulations mentioned and is a valuable addition to the suite of models/configurations simulating the Labrador Sea, but you should make these more clear. Please refer to the specific comments below for more details on this suggestion!

3. The current speed, convective energy, mixed layer depth, etc. are all valuable prop-

erties to investigate and clearly show the differences between the different horizontal resolutions for the processes of interest. However, considering this is the description paper of the model experiment, I suggest to add some analyses of temperature and salinity. Depth sections of temperature and salinity along AR7W for example could be compared to observations and should be familiar to most readers, thus providing a valuable reference future studies could compare their simulations to. Additionally this would give the reader some insight into the vertical structure of the simulation, as most quantities shown in the manuscript are surface values or depth-integrated.

4. There are many, many sentences starting with "While ...". This is not incorrect, however the readability would be greatly increased by a bit more work on the language. Additionally you could put more effort in smooth transitions between paragraphs, especially in the introduction.

5. There are hardly any specific numbers given in the manuscript. The properties are often described as being "large" or "smaller" etc. This makes the results hardly reproducible and also very hard to compare to other studies/models, especially given the fact, that the color scales of the figures are continuous and it is hard to read any values from the figures. Please refer to the specific comments for more details on this remark.

**Specific comments**

*Abstract*

*l. 10:* The restratification after convection could also be mentioned here, as this is a process that is expected to be differently resolved depending on the horizontal resolution.

*ll. 11-12 "We implemented [...]":* As you mentioned the 1/60° domain of the Labrador Sea before, this sentence reads as if they implemented additional nests into the 1/60° domain.

*l. 12 "[...] spans over 10 years":* See general comments above. (At least at the time of submission, this was not true.)

*l. 16:* Maybe better: "[...] impacts the simulation of the Labrador Sea." or "[...] impacts the representation of the Labrador Sea in the model."

**Introduction**

*ll. 23-25:* Confusing, first you describe that the current flows northwards and then you say it combines to be the WGC. In my opinion descriptions of current systems should be successively downstream, otherwise it is very hard to follow.

*ll. 28-29 "[...] now called the Labrador Current [...]":* Please specifiy from which point on the current is called Labrador Current. This is not clear to me at this point.

*ll. 31-42:* The paragraph describing the eddies in the Labrador Sea is a bit short, considering that resolving eddies is the major improvement and advantage of very high-resolution configurations. You could for example mention the ongoing debate over which type of eddies in the Labrador Sea is most important for the restratification and how your new simulation could help in solving this issue.

*l. 33:* "[...] instabilities that occur **within the boundary currents** along the shelf break." The boundary currents are a quite substantial ingredient to the instabilities and should be mentioned.

*ll. 41-42:* This statement requires a reference as it is not obvious that eddies generated at a western boundary should travel eastward into the basin.

*ll. 43-47:* Listing all the sites of deep convection seems unnecessary here, as they are never referred to again.

*ll. 48-49:* Weak stratification is a criterion for deep convection, the cyclonic circulation is not strictly necessary, please clarify this. For example "[...] weak stratification which is often achieved by a cyclonic circulation, [...]'".

*l. 53 "[...] relatively drier air [...]":* Relatively drier compared to what?

*l. 59:* You could mention the role of convective eddies in the restratification process

(Lilly et al., 2003; Rieck et al., 2019).

*ll. 60-62:* I do not understand what you want to convey with this sentence. Please clarify this, maybe you should consider the next sentence when rephrasing, as it seems that there is a repetition of the information that there is buoyant water transported towards the interior.

*ll. 67-68:* It is not clear what "throughout the North Altantic" refers to. The DWBC is part of the North Atlantic as well, but is mentioned seperately.

*l. 74:* You should briefly describe how polar amplification is causing additional freshwater and preferrably also include a reference for this statement.

*l. 78:* What kind of information does satellite altimetry provide and what is it used for?

*l. 87:* "larger spatial extent" compared to what?

*ll. 90-92:* One ofyour goals defined at the beginning is to investigate the role of horizontal resolution in simulations of the Labrador Sea. Here, it now seems as if this question has already been answered by an earlier study. You should make clear how your simulation is different to the earlier ones and how it can help in solving the question of how resolving eddies affects the Labrador Sea. Additionally you do not show any investigations into the numerical drift of your simulation, despite stating that the numerical drift is a major problem of simulations of the Labrador Sea even at high resolution.

*ll. 96-97:* You should carefully rephrase this sentence. Stating that the multi-decade $1/20°$ simulation resolves eddies in the Labrador Sea makes it hard to justify the need for a $1/60°$ simulation.

*ll. 101-102:* Instead of vaguely stating that these simulations have a length of "perhaps only a few years", you could exactly state how long the existing very high-resolution simulations of the Labrador Sea are.

*ll. 105-115:* See general comment 1.

***Methods***

*ll. 122-123:* I suggest briefly describing the extent of the domain here and use the figure as an additional source of information and not the only source.

*ll. 125-126:* I suggest briefly describing the extent of the domain here and use the figure as an additional source of information and not the only source.

*l. 124:* The mansucript would benefit from a short explanation of ARGIF's concept of parent and child domains at this point.

*l. 134:* You should cite Barnier et al. (2006) at this point.

*l. 144:* I am not sure whether you should phrase this as the "usual NEMO method". There are many ways to compute the mixed layer depth implemented in NEMO.

*l. 150:* Stating that "[...] smoothing between domains occurred [...]" makes it sound like you only had a passive role in that. It should be made clear, that you actively decided to smooth between domains.

*ll. 152-153:* Where and how exactly are the boundary conditions applied? Is there a sponge layer? Etc.

*ll. 155-159:* It seems random that you give a detailed explanation of how not explicitly including icebergs in the model could affect the freshwater budget of the subpolar North Atlantic, but do not mention other factors, like the choice of initial fields that could also significantly influence the freshwater budget.

*l. 160 "[...] including [...]":* Is there any part of the atmospheric forcing that is not listed afterwards? If not, then "including" is redundant.

*ll. 169-170:* What period does "long-term" refer to here and over which area is the heat loss calculated? These numbers should be made reproducible for comparison with future studies/forcing datasets/model simulations.

*ll. 174-175:* You never declared your interest in conducting a simulation past 2017, so the need for a different forcing set is not obvious here. Your goal to have a simulation that extends (almost) until present should be stated as it could be quite important for potential collaborators on the analysis of the produced model output!

*ll. 177-179:* I guess that the lack of interannual variability is caused by the missing deep convection due to the weak forcing. You should clarify this.

**[GMDD](https://www.geosci-model-dev-discuss.net/)**

Interactive
comment

***ll. 179-182:*** Do the different spatial and temporal resolutions have any consequences for the observed behaviour in Fig. 1? Otherwise I suggest to move this information to the part were the forcing datasets are described in general.

***ll. 182-185:*** You state that the remainder of the manuscript will only deal with LAB60-DFS, however later in the Methods section you describe the start of the computation at the Graham cluster and the spin-up period, which were done under CGRF atmospheric forcing if I understand correctly. This is confusing. I understand that describing such a complicated simulation pathway (switching forcing, switching computing cluster) is not an easy task. However, you should try to make it more clear. Is the whole used simulation from 2002-2011 called LAB60-DFS, or only the part from 2007-2011 where the DFS forcing is actually used?

***l. 186:*** It is not clear to me what "internal testing" refers to in this context. Why is it internal?

***l. 187:*** You should specify what "overhead" means. I guess it refers to the additional computational costs/time, but this needs to be made clear.

***ll. 187-193:*** A list with three items (the three passive tracers) would be a beneficial structural element at this point. Additionally, you should explain the choices made regarding the thresholds for the definition of the different water masses.

***ll. 194-195:*** You should clarify what "here" refers to, also there seems to be word missing between "such" and "resolution".

***ll. 208-209:*** It is not clear to me how you learned that your simulation is unstable from interpolating data. Additionally, "quickly go unstable" is rather unspecific.

***l. 211:*** You probably mean the opposite: Large (long) time step first, then decreasing to small (short) time step. Please clarifiy.

***ll. 217:*** At the end of the Methods section, the mansucript would greatly benefit from a clear short description of the simulation used for the analysis, making clear that the spin-up was done under CGRF forcing from 2002 to 2003, then there are 2004-2006 under CGRF forcing, followed by 2007-???? under DFS forcing. Maybe a simple schematic could help here, otherwise the reader has to skip back and forth through

the Methods section to gather this information.

**Model Simulation Results**

***ll. 220-222:*** Compared to what do the large differences occur? If you refer to the differences between LAB60, SPG12, and ANHA4, then it sounds like you want to use the large differences (compare the simulations) to understand the large differences. This sentence should be rephrased for clarification.

***ll. 224-225:*** At which depth are the current speeds compared? Additionally, you should provide some numbers here, "greater" and "slower" are not very specific.

***l. 230:*** Which shelf breaks? Shelf breaks have not been mentioned before in the manuscript.

***l. 231:*** At which depth is the eddy kinetic energy investigated?

***l. 231-253:*** You should use values when comparing the EKE in different regions and among different simulations. Using mostly larger and smaller makes it hard to keep track of how which result compares to which. Comparing numbers makes this a lot easier in most cases (additionally a comparison to other simulations and/or observations could be enabled).

***l. 231-232:*** You should clarify how you compute geostrophic velocities, and discuss that (at 1/60° resolution) your simulation might resolve submesoscale processes and features that are important for the restratification (among others) and are not completely represented by geostrophic currents.

***l. 232:*** In the definition of EKE, you should specifiy what the primes and overbars denote. Specifically, over which period the currents are averaged to calculate the deviations from. The choice of this period can have an influence on the results (Kang and Curchitser, 2017).

***l. 232:*** AVISO has not been introduced to the reader at this point. You should describe the data you use in the Methods section. Which data is used from AVISO, SSH or the geostrophic currents? Also, please note that many of the commonly

used SSH products are not distributed by AVISO anymore (since 2017) and it might be useful to update the data and use the new versions distributed by CMEMS (https://marine.copernicus.eu).

*l. 233 "[...] EKE coming from [...]":* One can not necessarily infer any direction of propagation from the maps of EKE. In these cases where it is likely that the high levels of EKE in one region are caused by propagation of (mostly) eddies into that region, I would suggest writing something along the lines of : "High levels of EKE can be found along the west coast of Greenland, extending into the interior of the basin at ?? North...". You should check the whole manuscript for this formulation ("[...] EKE coming from [...]") and adjust it.

*l. 235-236:* The result, that EKE is closely bound to the Labrador Current and the shelf break in the western Labrador Sea does not receive enough attention in my opinion. As far as I know, whether these boundary current eddies impact the deep convection region and the restratification or stay too close to the basin's boundary is still a matter of ongoing debate (e.g. Chanut et al., 2008; Gelderloos et al., 2011; Rieck et al., 2019) and a 1/60° simulation could clearly help in sovling this issue.

*l. 236 "[...] has lower levels of EKE [...]":* Lower compared to what?

*l. 238-240:* See above. EKE "coming from" somewhere and "entering" does not seem to be the best way do describe these results.

*l. 242-243:* See above. You could mention at some point earlier in the manuscript that "the EKE coming from the west coast of Greenland" is related to, or mostly consists of, Irminger Rings. That would probably help in describing the results later on (e.g. by using the phrase "Irminger Ring path" or something similar).

*l. 244 "AVISO observations":* See comment above on AVISO.

*l. 246-253:* I suggest that you additionally investigate EKE at depth. EKE at depth in the central Labrador Sea could be an indicator of the presence of convective eddies (which should be resolved at this resolution in contrast to other simulations with up to 1/20°) and could even be compared to observation of Fischer et al. (2018).

*l. 254:* I suggest you use a different date to show a snapshot from, as you mention

earlier that 2003 is still in the spin-up phase and you will only present model results from 2004 ongoing. This is not consistent.

*l. 255-256:* Again, no numbers, just "very strong" and "reduced".

*l. 256:* A brief explanation of convective energy would be extremely helpful here. The meaning of convective energy does not coincide with what the reader might intuitively think of when reading "convective energy" (the energy of convection).

*l. 261-264:* If you observe the described properties and processes in your simulation, I strongly suggest to show that and not just speculate. Additionally, there could be observational support for these speculations in Lilly et al. (2003) so I suggest checking that.

*ll. 276-277:* From the sentence, it is not clear to me whether the path of strong stratification is located between the 2500m and 3000m isobaths everywhere, or the path starts at the coastline between these isobaths. I suggest to formulate this more clearly.

*ll. 280-281:* This sentence sounds like the small convective energy is required to achieve weak stratification, whereas the convective energy is basically just another measure of stratification. I suggest rephrasing this. Addtionally, this fact has already been mentioned before.

*ll. 285-286 "[...], limiting the mixed depth between the 2000m and 3000m isobath.":* It is not clear to me what exactly you want to convey with this part of the sentence.

*l. 287:* You probably mean that the region where the mixed layer is deeper than a certain threshold is larger. The ocean has a mixed layer everywhere, so you cannot really reduce its spatial extent.

*l. 293:* The bottom of the mixed layer returns to the near-surface, the mixed layer is always connected to the surface.

*ll. 295-296:* There are several questions regarding the definition of the Labrador Sea Water. 1. How can you define the maximum density as the thickest depth? What is a "thickest depth"? 2. Referred to what does the density need to change by 0.001

$kg/m^3$? 3. Why do you calculate the MLD based on gradients and then for the definition of the LSW you use thresholds, wouldn't it be more consistent to also use a threshold for the MLD then? 4. How does this way of defining LSW compare to the way you defined your LSW tracer and what implications does this have?

**ll. 297-298:** Where could stair-stepping patterns emerge? Between years? And what are stair-stepping patterns? The same as staircase patterns?

**l. 316:** I am not completely sure what "enters the interior 2000m and 3000m isobath" means. Do you want to convey that the water mass propagates into regions where the water is between 2000 and 3000m deep? Please rephrase this sentence to make this more clear.

**ll. 318-319:** Stating that water ends up in the Labrador Current sounds like this water will never leave the Labrador Current. However, I suspect that the water still in the Labrador Current has just not yet left the current to the South or East due to the short integration time of your simulation. Using the phrase "ends up" is thus rather misleading.

**l. 323:** I am not sure what "within the 2000m and 3000m isobath" means.

**l. 324:** What is a "thicker amount"? Do you mean a "larger amount"?

**ll. 328-329:** Could you state how your definition of the tracer compares to your earlier definition of LSW?

**Discussion**

**ll. 337-339:** In the manuscript you do not really describe how submesoscale processes impact deep convection so it is irritating that you mention it in the discussion. I suggest that you add a paragraph to the Results section briefly showing that your simulation resolves the submesoscale and how that could impact deep convection and water mass formation. One of the key reasons to carry out a 1/60° simulation probably is that it resolves the mesoscale in the Labrador Sea and starts to resolve the larger end of the submesoscale range. I think you should make it more clear that

your simulation is capable of doing this and not just showing the end result (LSW for example) and speculate that the differences to lower resolutions are due to the missing (sub-)mesoscale.

*II. 348-349:* At this point you should compare your results from the 1/60° simulation to earlier studies with lower resolutions to point out the differences and espcecially improvements achieved by increasing resolution. At least for the Greenland meltwater, there are several studies investigating the fate of this tracer in simulations with lower resolutions (e.g. Böning et al., 2016 and others...).

**Tables**

*I. 526:* In the Methids section you state that you refer to the whole configuration is LAB60. In this table it looks like you refer to the parent domain as ANHA4, the first nest as SPG12 and only the second nest as LAB60. This should be made consistent.

**Figures**

In general I suggest to use larger fonts in the figures, especially for the titles (The titles should be at least as large as the manuscript font size.). Additionally, you could use some summarizing titles stating the property to be seen in the individual subplots (additional to the LAB60/SPG12/ANHA4/AVISO titles). I strongly recommend adding the units to the colorbars and also suggest using different colorscales, as these continuous scales sometime make it nearly impossible to read accurate values from the figure. It is not easy for example to distinguish between values of 0.2 $m/s$ and 0.3 $m/s$ in Figure 3 or 200 $cm^2/s^2$ and 500 $cm^2/s^2$ in Figure 4. (The colorscale used for the supplementary video showing LAB60s MLD is a good example of a discrete color scale where one can read values from the plots easier!)

*I. 552:* Speed at which depth?

*l. 557:* Eddy kinetic energy at which depth?
*l. 563:* Relative vortcity at which depth?
*l. 568:* Speed at which depth?

**Technical corrections**

*Abstract*

*ll. 9-10 "The transport of these fluxes [...]":* Transport and fluxes are used synony-
mously here and thus this should just read "These fluxes [...]" or "This transport [...]".

*Introduction*

*ll. 32:* Frajka-Williams
*ll. 80-83:* This sentence should be split for better readability.
*l. 89:* "[...], both which" should be rephrased
*l. 105:* "high resolution" should be "high-resolution"

*Methods*

*l. 123 "includes a nest":* To be precise, it includes two nests.
*l. 133 "horizontal grid resolution":* I suggest using "horizontal grid spacing" here.
*l. 136 "[...] primarily only [...]":* You should decide on either "primarily" or "only".
*l. 137-139 "All domains used [...]", "Lateral diffusion used [...]", etc.:* This should
be rephrased to something like "[...] scheme was used in all domains.", "A Laplacian
operator was used/implemented to compute lateral diffusion [...]", etc.
*l. 151:* "boundary nests" should be "the nest boundaries".
*l. 171:* "which were" should be "which was".
*l. 175 "[...] Fig. 1 identifies [...] between [...]":* should be "[...] Fig. 1 depicts [...] the

difference in mixed layer depth between [...]".

*l. 190:* "pathways which" should be "pathways along which".

*l. 194:* "masses" should be "water masses".

*l. 194:* "before in the past" should be either "before" or "in the past".

*l. 202:* "increase in simulation length" should probably be "decrease in simulation length".

*l. 207:* "[...] the occurrence of seasonal sea ice."

**Model Simulation Results**

*l. 222:* "ANHA12" should be "SPG12".

*l. 243:* "produce" should be "produced".

*l. 244:* "they match" should be "it matches".

*ll. 252-253:* duplicate mention of "supplemental"/"supplementary".

*l. 255:* "show" should be "shows".

*l. 270:* "ANHA12" should be "SPG12".

*l. 271:* "supplies" should be "supply".

*ll. 275-276:* duplicate use of "visible".

*l. 283:* "depth" should be "depths".

*l. 283:* "observation" should be "observations".

*ll. 299-301:* Please rephrase this sentence, the "though" seems unnecessary and "has this [...] being less dense." does not seem right.

*l. 305:* "between the" should be "in all three".

*l. 306:* "indicate that deep mixing is easier" should be something like "indicates that deep mixing is more likely".

**Discussion**

*l. 350:* "project" should be "projects".
**References**

**Barnier**, B., and Coauthors: Impact of partial steps and momentum advection schemes in a global ocean circulation model at eddy-permitting resolution. Ocean Dynamics, 56, 543–567, 2006. DOI: 10.1007/s10236-006-0082-1

**Böning**, C. W., Behrens, E., Biastoch, A., Getzlaff, K., and Bamber, J. L.: Emerging impact of Greenland meltwater on deepwater formation in the North Atlantic Ocean. Nature Geoscience, 9, 523-528. DOI: 10.1038/NGEO2740

**Chanut**, J., Barnier, B., Large, W., Debreu, L., Penduff, T., Molines, J.M., and Mathiot, P.: Mesoscale eddies in the Labrador Sea and their contribution to convection and restratification. Journal of Physical Oceanography, 28(8), 1617-1643, 2008.

**Fischer**, J., Karstensen, J., Oltmanns, M., and Schmidtko, S.: Mean circulation and EKE distribution in the Labrador Sea Water level of the subpolar North Atlantic. Ocean Sciences, 14, 1167-1183, 2018. DOI: 10.5194/os-14-1167-2018

**Fresnay**, S., Ponte, A. L., Le Gentil, S., Le Sommer, J.: Reconstruction of the 3-D dynamics from surface variable in a high-resolution simulation of the North Atlantic. Journal of Geophysical Research: Oceans, 123(3), 1612-1630, 2018.

**Gelderloos**, R., Katsman, C.A. and Drijfhout, S.S.: Assessing the roles of three eddy types in restratifying the Labrador Sea after deep convection. Journal of Physical Oceanography, 41(11), 2102-2119, 2011.

**Kang**, D., and Curchitser, E. N.: On the Evaluation of Seasonal Variability of the Ocean Kinetic Energy. Journal of Physical Oceanography, 47, 1675-1683, 2017. DOI: 10.1175/JPO-D-17-0063.1

**Lilly**, J.M., Rhines, P.B., Schott, F., Lavender, K., Lazier, J., Send, U., and D'Asaro, E.: Observations of the Labrador Sea eddy field. Progress in Oceanography, 59(1), 75-176, 2003.

**Rieck**, J. K., Böning, C. W., and Getzlaff, K.: The nature of eddy kinetic energy in the Labrador Sea: Different types of mesoscale eddies, their temporal variability, and

impact on deep convection. Journal of Physical Oceanography, 49(8), 2075-2094, 2019.

**Schubert**, R. Schwarzkopf, F. U., Baschek, B., Biastoch, A.: Submesoscale impacts on mesoscale Agulhas dynamics. Journal of Advances in Modeling Earth Systems, 11, 2019. DOI: 10.1029/2019MS001724

―――――――――――――

---

## Author Comment (AC1) · 1 Aug 2020

We (the authors) present the original responses in **bold** font while author responses will be in regular font. We only repost referee comments which contain suggestions.

Clark Pennelly and Paul G. Myers

**Referee comment #1 (Anonymous)**

**2 Specific comments**

**62-66: In this sentence, I find it difficult that you 1) use model results only to explain a real world phenomenon and 2) cite yourself only.**

We have included an additional 2 references, one using model results and the other an observational study. These studies investigated freshwater that enters the interior Labrador Sea, discussing freshwater pulses and their likely exchange from shelf to deep basin (observational study, Schmidt and Send 2007) or freshwater which leaves the Labrador Current (model study, McGeehan and Maslowski, 2011)

**73-77: There are other model studies showing the opposite, see e.g. Cael and Jansen (2020) and references therein.**

Added 2 citations (Cael and Jansen 2020 as well as Latif et al 2000) and text that discuss that while freshwater addition local to the Labrador Sea reduces convection and AMOC strength, freshwater addition that is non-local to the Labrador Sea drives the opposite.

**130: It is not discussed at all why a factor of 5 is used to obtain the 1/60◦ horizontal resolution. Please add this.**

We state that the factor of 5 is used to change the resolution from 1/12 to 1/60

**Fig. 3 and 4: The LAB60 North Atlantic Current seems to be less vivid and eddy-rich compared to both ANHA12 and AVISO. Can you discuss this?**

We add some discussion about why the North Atlantic Current seems less vivid and eddy-rich. We suspect it might have to do with the nested boundary, but other aspects of this simulation close to the remaining boundaries appear fine. We have later plans to investigate downstream influences of the LAB60 simulation in the SPG12 and ANHA4 domain.

**The "Discussion" section is rather a summary than a discussion. A discussion section is not explicitly required (https://www.geoscientific-model-development.net/for_authors/manuscript_preparation.html → Manuscript composition), so either rename the section or add a discussion. Personally, I would like to see a discussion. For instance, the videos indicate that the LAB60 setup exhibits a model drift and is far from equilibrium (drag the slider of the video player with**

the mouse from start to end rather fast and you can see a large-scale accumulation of the runoff and Irminger Water tracers in the respective videos). Would a potential model drift influence the LSW time series or the described eddy dynamics? In addition, the shown model data is particularly suited to discuss ongoing questions about meso- and submesoscale energy transfers during convective/unstable situations. Furthermore, your results indicate that even if an ocean model performs under a high spatial resolution, buoyant water needs to be provided by e.g. the boundary current in the first place to be available for eddies. If this is the case, a nested model configuration like the presented one would have a severe handicap given the coarser resolutions of the parents which provide the boundary current.

We have added 2 paragraphs to the discussion section that discuss model drift, Labrador Sea Water, potential boundary issues as well as others who have ran simulations of varying resolution within the Labrador Sea, particularly with the passive tracers as we use. We show that the model drift is significantly smaller in this configuration compared to ¼ and 1/12 degree via a new figure.

**3 Technical corrections**

**• sections are not numbered**

Sections are now numbered

**• its de Steur et al. 2009 (not 2018), Treguier (not Trequier) and Yashayaev (not Yashauaev); Fi* references not in alphabetical order**

We have corrected these (and some other) references/citations

**• 125 & 130: Please specify "temporal refinement".**

We help describe what 'temporal refinement' means by including the time step for all domains in this section.

**• Table 2: Please reference LIM2 and CORE as you did for other settings.**

We have included CGRF and DFS into this table. LIM2 and CORE have references here now as well.

**• Table 2: Please add the atmospheric forcings (CGRF and DFS) and their respective time periods**

See above

**• 160-185: LAB60 was forced by CGRF from 2002-2006 (5 years) and by DFS from 2007-2011 (5 years). If this is correct, I find it difficult to plot one LAB60 time series in Fig. 9. Can you at least indicate the two different forcing data sets in the plot/caption?**

We have made a new figure (Figure 2, below) that better explains the breakup of the LAB60 simulation with both the CGRF and DFS5.2 atmospheric forcing products.

[Figure]

• **189 & 329: Please specify which sigma/density is used.**

We state the density used: potential density

• **202: Did the simulation length really increase or rather decrease when the number of CPUs increased?**

We have better described that an increase in CPUs increased the number of simulated days per job submission on the high performance computing system.

• **225: configuration"s"?**

We have better clarified that the lower resolution simulations we compare against LAB60 are different configurations.

• **232: Please define how you computed the eddy components in the model and AVISO data.**

We have better defined how we calculate the eddy components in our EKE equation, both for the model and AVISO data.

• **Fig. 5 and Video 1: It is more informative to show relative vorticity normalized by the planetary vorticity, ζ/f, to learn about the transition from meso- to submesoscales.**

We have normalized vorticity against the planetary vorticity. However, little difference is noted across the new figure/video.

• **250 & 265: Please define meso- and sub-mesoscales in the introduction and how you separate them.**

We define mesoscale and sub-mesoscale in the introduction and how we separate them (by radius)

**• 256: Please define convective energy (CE) including the mixing depth to which you refer to in Fig. 6 b. On this snapshot, all CE values are >= 500 J m−3 . However, a winter situation is shown (17 January 2003) and I would expect CE values < 0 J m−3 indicating unstable situations. Is this a misunderstanding?**

We have included the reference depth in our definition now, 2000m. On the original Fig. 6b snapshot (which has since changed to 26 July 2007 as per Reviewer 2's comment), CE values would only be 0 if there existed no potential density change between the surface and 2000m. LAB60 does not have a mixed layer which reaches that deep (see Fig. 9) and thus always has positive CE. Furthermore, values less than 0 do not occur as the model's vertical mixing scheme quickly deals with unstable vertical situations.

**• 295: First, you find the LSW with σΘ > 27.68 kg m−3 (line 189; I assume you used σΘ in this step). Then, you calculate σ1 of this water mass. Then, I don't understand why you define the "yearly maximum density of this water mass as the thickest depth where the density changes by 0.001 kg m−3 ". I don't understand how a "depth" can be "thick" and to which density the 0.001 kg m−3 change refers to. Can you reformulate this here and in the caption of Fig. 9? Formulated as it is, I would have problems reproducing this quantity.**

We have completely rewritten (but not changed the calculation) the description of how we determine LSW layer thickness

**• Videos: Adding the sea ice edge and some MLD contours would make the videos even more helpful.**

We have added a sea-ice edge and a 1000 m MLD contour in the MLD video. We have also updated all videos to the end of 2013.

**Referee comment #2 (Jan Klaus Rieck)**

**General comments**

**1. You emphasize the need for a long simulation to study the decadal variability of the Labrador Sea and state that your simulation is suitable for that. However, at least at the time of submission, there were only 7 years (excluding spin-up) of the simulation finished and your statement of having a simulation of more than ten years do not hold. If the simulation does now (time of revision) extend over the stated period, this is fine, if not there need to be adjustments to the manuscript. Additionally, you should think about rephrasing the manuscript at some points concerning the suitability of a ~10 year-long simulation to study decadal variability.**

We have changed the writing of the manuscript to shift away from decadal variability to interannual variability. Additionally, we mention, and show (figures/movies) data from 2004 through 2013, 3 additional years since our original submission.

**2. In my eyes, you do not satisfactorily advertise the advantages and improvements of LAB60 in comparison to other, existing high-resolution simulations carried out with the same model (e.g. the VIKING20X simulations at 1/20◦ [Rieck et al., 2019]). I do think that the presented LAB60 simulation has advantages over the other simulations mentioned and is a valuable addition to the suite of models/configurations simulating the Labrador Sea, but you should make these more clear. Please refer to the specific comments below for more details on this suggestion!**

We have added additional text that should better illustrate how LAB60 offer advantages over other high resolution simulations in the Labrador Sea. We discuss how our simulation is much longer in simulation length than the other 1/60 simulations and includes passive tracers which high-resolution simulations often do not have.

**3. The current speed, convective energy, mixed layer depth, etc. are all valuable properties to investigate and clearly show the differences between the different horizontal resolutions for the processes of interest. However, considering this is the description paper of the model experiment, I suggest to add some analyses of temperature and salinity. Depth sections of temperature and salinity along AR7W for example could be compared to observations and should be familiar to most readers, thus providing a valuable reference future studies could compare their simulations to. Additionally this would give the reader some insight into the vertical structure of the simulation, as most quantities shown in the manuscript are surface values or depth-integrated.**

We have included a T/S/density figure of LAB60 and observations across AR7W (below) as well as some text describing the model versus observations.

[Figure]

**4. There are many, many sentences starting with "While ...". This is not incorrect, however the readability would be greatly increased by a bit more work on the language. Additionally you could put more effort in smooth transitions between paragraphs, especially in the introduction.**

We have changed many of our transition sentences to increase readability as well as reduced our use of 'while'.

**5. There are hardly any specific numbers given in the manuscript. The properties are often described as being "large" or "smaller" etc. This makes the results hardly reproducible and also very hard to compare to other studies/models, especially given the fact, that the color scales of the figures are continuous and it is hard to read any values from the figures. Please refer to the specific comments for more details on this remark**

We have added specific numbers and/or range of values throughout the manuscript and removed vague descriptors. Colorbars in many figures have been changed to be easier to read.

**Specific comments**

**Abstract**

**l. 10: The restratification after convection could also be mentioned here, as this is a process that is expected to be differently resolved depending on the horizontal resolution.**

We include more writing about restratification that occurs after convection.

**ll. 11-12 "We implemented [...]": As you mentioned the 1/60◦ domain of the Labrador Sea before, this sentence reads as if they implemented additional nests into the 1/60◦ domain.**

Clarified how the nests are used in our configuration

**l. 12 "[...] spans over 10 years": See general comments above. (At least at the time of submission, this was not true.)**

We have clarified here (and in a few other locations) that our simulation is still being carried out. As of this revision, there is 10 yeas of non-spinup model output (2004-2013) and we will run the simulation with DFS forcing through the end of 2017 (when DFS ends). We will eventually swap to another forcing set, though it is still undecided which one we will use.

**l. 16: Maybe better: "[...] impacts the simulation of the Labrador Sea." or "[...] impacts the representation of the Labrador Sea in the model."**

Changed to make clearer

**Introduction**

**ll. 23-25: Confusing, first you describe that the current flows northwards and then you say it combines to be the WGC. In my opinion descriptions of current systems should be successively downstream, otherwise it is very hard to follow.**

Reordered the description of the currents to be downstream of one another.

**ll. 28-29 "[...] now called the Labrador Current [...]": Please specifiy from which point on the current is called Labrador Current. This is not clear to me at this point.**

We now state that the Labrador Current begins in proximity to Hudson Strait

**ll. 31-42: The paragraph describing the eddies in the Labrador Sea is a bit short, considering that resolving eddies is the major improvement and advantage of very high-resolution configurations. You**

**could for example mention the ongoing debate over which type of eddies in the Labrador Sea is most important for the restratification and how your new simulation could help in solving this issue.**

We have added a few more sentences that describes eddies in the Labrador Sea. We also mention the debate on which eddies influence the stratification.

**33: "[...] instabilities that occur within the boundary currents along the shelf break." The boundary currents are a quite substantial ingredient to the instabilities and should be mentioned.**

We have made this sentence clearer.

**ll. 41-42: This statement requires a reference as it is not obvious that eddies generated at a western boundary should travel eastward into the basin.**

We include a citation to a numerical modelling study (Pennelly et al., 2019) that shows the eddy fluxes from the WGC and Labrador Current provide a net flux towards the interior of the Labrador Sea.

**ll. 43-47: Listing all the sites of deep convection seems unnecessary here, as they are never referred to again.**

We have removed non-Labrador Sea convection locations since we do not refer back to them.

**ll. 48-49: Weak stratification is a criterion for deep convection, the cyclonic circulation is not strictly necessary, please clarify this. For example "[...] weak stratification which is often achieved by a cyclonic circulation, [...]"'.**

We have made it clearer that cyclonic circulation isn't a requirement but rather helps set the overall weak stratification required for deep convection

**l. 53 "[...] relatively drier air [...]": Relatively drier compared to what?**

Changed to 'dry air' as we are not referencing against anything at this point

**l. 59: You could mention the role of convective eddies in the restratification process (Lilly et al., 2003; Rieck et al., 2019).**

We added text which discusses the roles of convective eddies in the restratification process and used the suggested references.

**ll. 60-62: I do not understand what you want to convey with this sentence. Please clarify this, maybe you should consider the next sentence when rephrasing, as it seems that there is a repetition of the information that there is buoyant water transported towards the interior.**

We have rephrased this confusing sentence

**ll. 67-68: It is not clear what "throughout the North Altantic" refers to. The DWBC is part of the North Atlantic as well, but is mentioned seperately.**

We have cleared up this part regarding where newly formed Labrador Sea Water flows to.

**l. 74: You should briefly describe how polar amplification is causing additional freshwater and preferrably also include a reference for this statement.**

We briefly describe polar amplification in relation to ice-albedo feedback loop and how the additional melt enters the boundary currents.

**l. 78: What kind of information does satellite altimetry provide and what is it used for?**

We add some text regarding satellite altimetry data.

**l. 87: "larger spatial extent" compared to what?**

We clarified what we originally meant by 'larger spatial extent'

**ll. 90-92: One of your goals defined at the beginning is to investigate the role of horizontal resolution in simulations of the Labrador Sea. Here, it now seems as if this question has already been answered by an earlier study. You should make clear how your simulation is different to the earlier ones and how it can help in solving the question of how resolving eddies affects the Labrador Sea. Additionally you do not show any investigations into the numerical drift of your simulation, despite stating that the numerical drift is a major problem of simulations of the Labrador Sea even at high resolution.**

We add discussion about how our simulation is useful in context to others. We include an additional figure (below) that shows numerical drift of the lower-resolution simulations as well as LAB60 (which has almost no drift).

[Figure]

**ll. 96-97: You should carefully rephrase this sentence. Stating that the multi-decade 1/20◦ simulation resolves eddies in the Labrador Sea makes it hard to justify the need for a 1/60◦ simulation.**

We have clarified that 1/20 degree simulations may be lacking or misrepresenting the submesoscale that higher-resolution simulations are needed for.

**ll. 101-102: Instead of vaguely stating that these simulations have a length of "perhaps only a few years", you could exactly state how long the existing very high-resolution simulations of the Labrador Sea are.**

We have added information on roughly how long these high-resolution simulations are

**ll. 105-115: See general comment 1.**

We have edited these lines so that we are writing about our LAB60 simulation in regards to interannual variability rather than decadal variability.

**Methods**

**ll. 122-123: I suggest briefly describing the extent of the domain here and use the figure as an additional source of information and not the only source.**

We have added a brief description to the ANHA4 domain but not the SPG12 or LAB60 spatial extent of their domain. Figure 1 shows the spatial extent of each domain

**ll. 125-126: I suggest briefly describing the extent of the domain here and use the figure as an additional source of information and not the only source.**

Same as above

**l. 124: The mansucript would benefit from a short explanation of ARGIF's concept of parent and child domains at this point.**

We have added some text that described briefly how AGRIF treats the boundary conditions between parent and nest.

**l. 134: You should cite Barnier et al. (2006) at this point.**

We have included a citation to Barnier et al 2006 in regards to partial cells

**l. 144: I am not sure whether you should phrase this as the "usual NEMO method". There are many ways to compute the mixed layer depth implemented in NEMO.**

We have clarified this sentence so the reader understands that it was our usual methods of calculating the MLD, not the only one.

**l. 150: Stating that "[...] smoothing between domains occurred [...]" makes it sound like you only had a passive role in that. It should be made clear, that you actively decided to smooth between domains.**

We have made it more clear what we meant by bathymetric smoothing along the nested boundaries.

**ll. 152-153: Where and how exactly are the boundary conditions applied? Is there a sponge layer? Etc.**

We state how and where the boundary conditions are applied as well as the lack of a sponge layer.

**ll. 155-159: It seems random that you give a detailed explanation of how not explicitly including icebergs in the model could affect the freshwater budget of the subpolar North Atlantic, but do not mention other factors, like the choice of initial fields that could also significantly influence the freshwater budget.**

We have removed some text regarding freshwater from icebergs. We did keep text that states AGRIF doesn't work with the iceberg model, primarily to state that we turned solid freshwater into liquid freshwater and used the same volume.

**l. 160 "[...] including [...]": Is there any part of the atmospheric forcing that is not listed afterwards? If not, then "including" is redundant.**

Removed 'including'

**ll. 169-170: What period does "long-term" refer to here and over which area is the heat loss calculated? These numbers should be made reproducible for comparison with future studies/forcing datasets/model simulations.**

We clarify the period which 'long-term' refers to.

**ll. 174-175: You never declared your interest in conducting a simulation past 2017, so the need for a different forcing set is not obvious here. Your goal to have a simulation that extends (almost) until present should be stated as it could be quite important for potential collaborators on the analysis of the produced model output!**

We clarify here, and in other locations, that we intend to not only drive this simulation up to present time (currently it is in 2015), but to keep it at the near-present time by using recently released forcing.

**ll. 177-179: I guess that the lack of interannual variability is caused by the missing deep convection due to the weak forcing. You should clarify this.**

We clarify that the LAB60-CGRF is missing deep convection after a certain point which we attribute it to weaker forcing.

**ll. 179-182: Do the different spatial and temporal resolutions have any consequences for the observed behaviour in Fig. 1? Otherwise I suggest to move this information to the part were the forcing datasets are described in general.**

We migrate the information on the atmospheric forcing resolution from here to the area where we discuss these datasets.

**ll. 182-185: You state that the remainder of the manuscript will only deal with LAB60-DFS, however later in the Methods section you describe the start of the computation at the Graham cluster and the spin-up period, which were done under CGRF atmospheric forcing if I understand correctly. This is confusing. I understand that describing such a complicated simulation pathway (switching forcing, switching computing cluster) is not an easy task. However, you should try to make it more clear. Is the whole used simulation from 2002-2011 called LAB60-DFS, or only the part from 2007-2011 where the DFS forcing is actually used?**

We have added a new figure that should quickly and easily describe the previously confusing sentences. See Figure 2.

**l. 186: It is not clear to me what "internal testing" refers to in this context. Why is it internal?**

We have made 'internal testing' a bit more clear by simply stating that it was during our 'early testing'

**l. 187: You should specify what "overhead" means. I guess it refers to the additional computational costs/time, but this needs to be made clear.**

We removed 'overheard' and instead state that each passive tracers takes additional computer resources

**ll. 187-193: A list with three items (the three passive tracers) would be a beneficial structural element at this point. Additionally, you should explain the choices made regarding the thresholds for the definition of the different water masses.**

We list the tracers in bullet point form as suggested

**ll. 194-195: You should clarify what "here" refers to, also there seems to be word missing between "such" and "resolution".**

We have clarified this sentence

**ll. 208-209: It is not clear to me how you learned that your simulation is unstable from interpolating data. Additionally, "quickly go unstable" is rather unspecific.**

We have written a bit more about what went wrong during the spinup period when instabilities built and crashed the model.

**l. 211: You probably mean the opposite: Large (long) time step first, then decreasing to small (short) time step. Please clarifiy.**

Our spinup process, in regards to how we changed the numerical time step, was written correctly. We started with a very short timestep (2 seconds) which we then increased over time to the final value (48s). We did clarify our spinup process paragraph significantly.

**ll. 217: At the end of the Methods section, the mansucript would greatly benefit from a clear short description of the simulation used for the analysis, making clear that the spin-up was done under**

**CGRF forcing from 2002 to 2003, then there are 2004-2006 under CGRF forcing, followed by 2007-????
under DFS forcing. Maybe a simple schematic could help here, otherwise the reader has to skip back
and forth through the Methods section to gather this information.**

Figure 2 was produced to make this transition from CGRF to DFS more clear.

**Model Simulation Results**

**ll. 220-222: Compared to what do the large differences occur? If you refer to the differences between
LAB60, SPG12, and ANHA4, then it sounds like you want to use the large differences (compare the
simulations) to understand the large differences. This sentence should be rephrased for clarification.**

We have made this sentence more clear

**ll. 224-225: At which depth are the current speeds compared? Additionally, you should provide some
numbers here, "greater" and "slower" are not very specific.**

We have added text to state the current speeds were calculated from the top 50m for the simulations.

**l. 230: Which shelf breaks? Shelf breaks have not been mentioned before in the manuscript.**

We mention which shelf breaks we are referring to: the Labrador coast and the western side of
Greenland

**l. 231: At which depth is the eddy kinetic energy investigated?**

We have clarified that EKE was calculated from geostrophic velocities derived from the sea level
anomaly

**l. 231-253: You should use values when comparing the EKE in different regions and among different
simulations. Using mostly larger and smaller makes it hard to keep track of how which result
compares to which. Comparing numbers makes this a lot easier in most cases (additionally a
comparison to other simulations and/or observations could be enabled).**

We include EKE values when comparing the different simulations.

**l. 231-232: You should clarify how you compute geostrophic velocities, and discuss that (at 1/60◦
resolution) your simulation might resolve submesoscale processes and features that are important for
the restratification (among others) and are not completely represented by geostrophic currents.**

We clarify how we compute geostrophic velocities.

**l. 232: In the definition of EKE, you should specifiy what the primes and overbars denote. Specifically,
over which period the currents are averaged to calculate the deviations from. The choice of this
period can have an influence on the results (Kang and Curchitser, 2017).**

See above

**l. 232: AVISO has not been introduced to the reader at this point. You should describe the data you use in the Methods section. Which data is used from AVISO, SSH or the geostrophic currents? Also, please note that many of the commonly used SSH products are not distributed by AVISO anymore (since 2017) and it might be useful to update the data and use the new versions distributed by CMEMS (https://marine.copernicus.eu).**

We now include some text regarding AVISO data earlier in the methods section, as well as which data we used from AVISO.

**l. 233 "[...] EKE coming from [...]": One can not necessarily infer any direction of propagation from the maps of EKE. In these cases where it is likely that the high levels of EKE in one region are caused by propagation of (mostly) eddies into that region, I would suggest writing something along the lines of : "High levels of EKE can be found along the west coast of Greenland, extending into the interior of the basin at ?? North...". You should check the whole manuscript for this formulation ("[...] EKE coming from [...]") and adjust it.**

We have changed many sentences which originally stated 'EKE came/coming from' to 'EKE extending' or something similar.

**l. 235-236: The result, that EKE is closely bound to the Labrador Current and the shelf break in the western Labrador Sea does not receive enough attention in my opinion. As far as I know, whether these boundary current eddies impact the deep convection region and the restratification or stay too close to the basin's boundary is still a matter of ongoing debate (e.g. Chanut et al., 2008; Gelderloos et al., 2011; Rieck et al., 2019) and a 1/60∘ simulation could clearly help in sovling this issue.**

We add more text regarding the lack of EKE extending from the Labrador Current into the interior Labrador Sea, suggesting that boundary current eddies here likely do not influence the stratification/restratification of the Labrador Sea.

**l. 236 "[...] has lower levels of EKE [...]": Lower compared to what?**

We have changed many sentences to indicate a range of EKE values rather than just 'lower/greater/etc'

**l. 238-240: See above. EKE "coming from" somewhere and "entering" does not seem to be the best way do describe these results.**

We have changed sentences with 'EKE coming/entering/etc' to be more clear and accurate.

**l. 242-243: See above. You could mention at some point earlier in the manuscript that "the EKE coming from the west coast of Greenland" is related to, or mostly consists of, Irminger Rings. That would probably help in describing the results later on (e.g. by using the phrase "Irminger Ring path" or something similar).**

We have changed this sentence to state that the EKE signature here is related to Irminger Rings

**l. 244 "AVISO observations": See comment above on AVISO.**

We have clarified our writing regarding AVISO observations

**l. 246-253: I suggest that you additionally investigate EKE at depth. EKE at depth in the central Labrador Sea could be an indicator of the presence of convective eddies (which should be resolved at this resolution in contrast to other simulations with up to 1/20° ) and could even be compared to observation of Fischer et al. (2018).**

We decided to not include any investigation of EKE at depth which could be evidence of convective eddies. We have future goals to look into these eddies from the results of this simulation.

**l. 254: I suggest you use a different date to show a snapshot from, as you mention earlier that 2003 is still in the spin-up phase and you will only present model results from 2004 ongoing. This is not consistent.**

We have changed the date of this figure to be outside the spinup/adjustment phase.

**l. 255-256: Again, no numbers, just "very strong" and "reduced".**

We have included numbers in place of 'very strong', 'reduced', and other vague descriptors.

**l. 256: A brief explanation of convective energy would be extremely helpful here. The meaning of convective energy does not coincide with what the reader might intuitively think of when reading "convective energy" (the energy of convection).**

We add further explanation on what convective energy means

**l. 261-264: If you observe the described properties and processes in your simulation, I strongly suggest to show that and not just speculate. Additionally, there could be observational support for these speculations in Lilly et al. (2003) so I suggest checking that.**

We include references to Lilly et al., 2003 for observational support on our speculation.

**ll. 276-277: From the sentence, it is not clear to me whether the path of strong stratification is located between the 2500m and 3000m isobaths everywhere, or the path starts at the coastline between these isobaths. I suggest to formulate this more clearly.**

We have made this sentence more clear

**ll. 280-281: This sentence sounds like the small convective energy is required to achieve weak stratification, whereas the convective energy is basically just another measure of stratification. I suggest rephrasing this. Addtionally, this fact has already been mentioned before.**

We have removed this as it was already mentioned earlier, which we also clarified.

**ll. 285-286 "[...], limiting the mixed depth between the 2000m and 3000m isobath.": It is not clear to me what exactly you want to convey with this part of the sentence.**

We have clarified this sentence in regards to the mixed layer

**l. 287: You probably mean that the region where the mixed layer is deeper than a certain threshold is larger. The ocean has a mixed layer everywhere, so you cannot really reduce its spatial extent.**

We have clarified this sentence

**l. 293: The bottom of the mixed layer returns to the near-surface, the mixed layer is always connected to the surface.**

We have clarified this sentence by stating the 'bottom of the mixed layer'

**ll. 295-296: There are several questions regarding the definition of the Labrador Sea Water. 1. How can you define the maximum density as the thickest depth? What is a "thickest depth"? 2. Referred to what does the density need to change by 0.001 kg/m3? 3. Why do you calculate the MLD based on gradients and then for the definition of the LSW you use thresholds, wouldn't it be more consistent to also use a threshold for the MLD then? 4. How does this way of defining LSW compare to the way you defined your LSW tracer and what implications does this have?**

We have clarified how we calculate Labrador Sea Water, these sentences have been completely rewritten in a manner that should make more sense. The calculation has not changed. In our introduction, we state that the thresholding of MLD doesn't work well for deep convection as T/S compensate with little change in density. We do state that our LSW tracer is very different than how we calculate LSW from the output of our model: we couldn't factor any potential drift while the tracers are being formed. The result is they show different aspects: The tracer shows newly formed Labrador Sea Water while our LSW calculations show how the thickness and density within our region of interest evolve in time. Both examine Labrador Sea Water in their own way.

**ll. 297-298: Where could stair-stepping patterns emerge? Between years? And what are stair-stepping patterns? The same as staircase patterns?**

We clarify what we meant by stair-stepping/staircase.

**l. 316: I am not completely sure what "enters the interior 2000m and 3000m isobath" means. Do you want to convey that the water mass propagates into regions where the water is between 2000 and 3000m deep? Please rephrase this sentence to make this more clear.**

We clarify this sentence regarding propagation into deeper water

**ll. 318-319: Stating that water ends up in the Labrador Current sounds like this water will never leave the Labrador Current. However, I suspect that the water still in the Labrador Current has just not yet left the current to the South or East due to the short integration time of your simulation. Using the phrase "ends up" is thus rather misleading.**

We change the phrasing of this sentence to be clearer

**l. 323: I am not sure what "within the 2000m and 3000m isobath" means.**

We clarify this sentence in regards to the different isobaths

**l. 324: What is a "thicker amount"? Do you mean a "larger amount"?**

We did mean a 'larger amount' and have fixed it

**ll. 328-329: Could you state how your definition of the tracer compares to your earlier definition of LSW?**

We added a sentence regarding the difference between our LSW tracer and our LSW calculation.

**Discussion**

**ll. 337-339: In the manuscript you do not really describe how submesoscale processes impact deep convection so it is irritating that you mention it in the discussion. I suggest that you add a paragraph to the Results section briefly showing that your simulation resolves the submesoscale and how that could impact deep convection and water mass formation. One of the key reasons to carry out a 1/60° simulation probably is that it resolves the mesoscale in the Labrador Sea and starts to resolve the larger end of the submesoscale range. I think you should make it more clear that your simulation is capable of doing this and not just showing the end result (LSW for example) and speculate that the differences to lower resolutions are due to the missing (sub-)mesoscale.**

We have added a bit more discussion in regards to mesoscale to submesoscale process here, though it is just text and not additional figures. Rather than include additional figures in this manuscript, we are currently writing up another manuscript which further investigates these processes in much more detail.

**ll. 348-349: At this point you should compare your results from the 1/60° simulation to earlier studies with lower resolutions to point out the differences and espcecially improvements achieved by increasing resolution. At least for the Greenland meltwater, there are several studies investigating the fate of this tracer in simulations with lower resolutions (e.g. Böning et al., 2016 and others...).**

We have added some comparison of our LAB60 simulation to earlier low-resolution studies, particularly ones with similar passive tracers as the ones we use

**Tables**

**l. 526: In the Methids section you state that you refer to the whole configuration is LAB60. In this table it looks like you refer to the parent domain as ANHA4, the first nest as SPG12 and only the second nest as LAB60. This should be made consistent.**

We have added additional text to try and make it more clear that while we call this configuration 'LAB60', the entire configuration contains 3 domains: ANHA4, SPG12 and LAB60.

**Figures**

**In general I suggest to use larger fonts in the figures, especially for the titles (The titles should be at least as large as the manuscript font size.). Additionally, you could use some summarizing titles stating the property to be seen in the individual subplots (additional to the LAB60/SPG12/ANHA4/AVISO titles). I strongly recommend adding the units to the colorbars and also suggest using different colorscales, as these continuous scales sometime make it nearly impossible to read accurate values from the figure. It is not easy for example to distinguish between values of 0.2 m/s and 0.3 m/s in Figure 3 or 200 cm2/s2 and 500 cm2/s2 in Figure 4. (The colorscale used for the supplementary video showing LAB60s MLD is a good example of a discrete color scale where one can read values from the plots easier!)**

We have remade most figures with larger font. Some figures have been updated so their titles are a bit more descriptive rather than just 'ANHA4', they now include the proxy being investigated (I.e. "ANHA4 MLD"). We have changed many colorbars to show more discrete features.

**l. 552: Speed at which depth?**

We now state the speed was calculated over the top 50m

**l. 557: Eddy kinetic energy at which depth?**

We now clarify that EKE was calculated via the geostrophic velocities resulting from the sea level anomaly.

**l. 563: Relative vortcity at which depth?**

We now state vorticity was calculated over the top 50m

**l. 568: Speed at which depth?**

We now state that speed here was just shown for the surface.

**Technical corrections**

**Abstract**

**ll. 9-10 "The transport of these fluxes [...]": Transport and fluxes are used synonymously here and thus this should just read "These fluxes [...]" or "This transport [...]".**

Done

**Introduction**

**ll. 32: Frajka-Williams**

Fixed the misspelling of Frajka-Williams

**ll. 80-83: This sentence should be split for better readability.**

We have made this sentence more readable

**l. 89: "[...], both which" should be rephrased**

We have changed this fragment to be more readable

**l. 105: "high resolution" should be "high-resolution"**

We have hyphenated this and many other instances where we missed it before

**Methods**

**l. 123 "includes a nest": To be precise, it includes two nests.**

We have changed our wording choice to be more precise.

**l. 133 "horizontal grid resolution": I suggest using "horizontal grid spacing" here.**

Changed

**l. 136 "[...] primarily only [...]": You should decide on either "primarily" or "only".**

Changed

**l. 137-139 "All domains used [...]", "Lateral diffusion used [...]", etc.: This should be rephrased to something like "[...] scheme was used in all domains.", "A Laplacian operator was used/implemented to compute lateral diffusion [...]", etc.**

We have reworded how we describe the various domains and their schemes

**l. 151: "boundary nests" should be "the nest boundaries".**

Changed

**l. 171: "which were" should be "which was".**

Changed

**l. 175 "[…] Fig. 1 identifies […] between […]": should be "[…] Fig. 1 depicts […] the difference in mixed layer depth between […]".**

Changed

**l. 190: "pathways which" should be "pathways along which".**

Changed

**l. 194: "masses" should be "water masses".**

Changed

**l. 194: "before in the past" should be either "before" or "in the past".**

Changed

**l. 202: "increase in simulation length" should probably be "decrease in simulation length".**

We have reworded this sentence to make it clearer that more CPUs allowed us to produce more simulated days in the same amount of requested job time.

**l. 207: "[…] the occurrence of seasonal sea ice."**

Changed

**Model Simulation Results**

**l. 222: "ANHA12" should be "SPG12".**

We did not make this change as we are purposely comparing the LAB60 configuration against the ANHA4 and ANHA12 configuration, not SPG12

**l. 243: "produce" should be "produced".**

Changed

**l. 244: "they match" should be "it matches".**

Changed

**ll. 252-253: duplicate mention of "supplemental"/"supplementary".**

Removed the extra mention of 'supplemental' here and in a few more instances

**l. 255: "show" should be "shows". l. 270: "ANHA12" should be "SPG12".**

Changed

**l. 271: "supplies" should be "supply".**

Changed

**ll. 275-276: duplicate use of "visible".**

Changed

**l. 283: "depth" should be "depths".**

Changed

**l. 283: "observation" should be "observations".**

Changed

**ll. 299-301: Please rephrase this sentence, the "though" seems unnecessary and "has this [...] being less dense." does not seem right.**

We have rewritten a large portion of this paragraph to make it more clear- this sentence has been corrected

**l. 305: "between the" should be "in all three".**

Changed

**l. 306: "indicate that deep mixing is easier" should be something like "indicates that deep mixing is more likely".**

Changed

**Discussion**

**l. 350: "project" should be "projects"**

Changed

**References**

Some of the following references were already included in our manuscript. Many others were added as suggested by this reviewer.

**Barnier, B., and Coauthors: Impact of partial steps and momentum advection schemes in a global ocean circulation model at eddy-permitting resolution. Ocean Dynamics, 56, 543–567, 2006. DOI: 10.1007/s10236-006-0082-1**

Böning, C. W., Behrens, E., Biastoch, A., Getzlaff, K., and Bamber, J. L.: Emerging impact of Greenland meltwater on deepwater formation in the North Atlantic Ocean. Nature Geoscience, 9, 523-528. DOI: 10.1038/NGEO2740

Chanut, J., Barnier, B., Large, W., Debreu, L., Penduff, T., Molines, J.M., and Mathiot, P.: Mesoscale eddies in the Labrador Sea and their contribution to convection and restratification. Journal of Physical Oceanography, 28(8), 1617-1643, 2008.

Fischer, J., Karstensen, J., Oltmanns, M., and Schmidtko, S.: Mean circulation and EKE distribution in the Labrador Sea Water level of the subpolar North Atlantic. Ocean Sciences, 14, 1167-1183, 2018. DOI: 10.5194/os-14-1167-2018

Fresnay, S., Ponte, A. L., Le Gentil, S., Le Sommer, J.: Reconstruction of the 3-D dynamics from surface variable in a high-resolution simulation of the North Atlantic. Journal of Geophysical Research: Oceans, 123(3), 1612-1630, 2018.

Gelderloos, R., Katsman, C.A. and Drijfhout, S.S.: Assessing the roles of three eddy types in restratifying the Labrador Sea after deep convection. Journal of Physical Oceanography, 41(11), 2102-2119, 2011.

Kang, D., and Curchitser, E. N.: On the Evaluation of Seasonal Variability of the Ocean Kinetic Energy. Journal of Physical Oceanography, 47, 1675-1683, 2017. DOI: 10.1175/JPO-D-17-0063.1

Lilly, J.M., Rhines, P.B., Schott, F., Lavender, K., Lazier, J., Send, U., and D'Asaro, E.: Observations of the Labrador Sea eddy field. Progress in Oceanography, 59(1), 75-176, 2003.

Rieck, J. K., Böning, C. W., and Getzlaff, K.: The nature of eddy kinetic energy in the Labrador Sea: Different types of mesoscale eddies, their temporal variability, and impact on deep convection. Journal of Physical Oceanography, 49(8), 2075-2094, 2019.

Schubert, R. Schwarzkopf, F. U., Baschek, B., Biastoch, A.: Submesoscale impacts on mesoscale Agulhas dynamics. Journal of Advances in Modeling Earth Systems, 11, 2019. DOI: 10.1029/2019MS001724

---

## Author Response (AR2)

We (the authors) present the original responses in **bold** font while author responses will be in regular font. We indicate the line number where text was added/modified according to the new manuscript. We will type the line number in bold (**L##**) following our response to the comment, while any text which was modified will be in red. Our marked manuscript follows.

Clark Pennelly and Paul G. Myers

**Reviewer 1**

**The authors have greatly improved the manuscript compared to the original submission. The advantages of the LAB60 simulation over other existing high-resolution simulations of the Labrador Sea have been made clear. Additional and revised text, figures and diagrams address the comments made by the reviewers before and thus there are only few remaining issues to be solved before publication as outlined below in the Specific Comments.**

**Specific Comments:**

**11-12: This sentence is not really more clear than before. It still fives the impression that your regional configuration is set up within the Labrador Sea and the nests are implemented therein.**

We have modified this sentence to "Our regional configuration, spanning the full North Atlantic and Arctic, includes nested domains within the North Atlantic and Labrador Sea, reducing computational costs that allow for a lengthy simulation from 2002 to the near-present time.". **L11-13**

**14-16: In the author response you state that you changed this sentence, however it is identical to the original manuscript.**

We have modified the end of this sentence to be a bit more clear, using your previous recommendation: "We describe the configuration setup and compare against similarly forced lower-resolution simulations to better describe how horizontal resolution impacts the representation of the Labrador Sea in the model.". **L17**

**168: "with" should probably be "and".**

Fixed; **L176**

**194:"configuration" should be "configurations".**

Fixed. **L203**

**283: "LS60" should be "LAB60".**

Fixed. **L294**

**352: "math far closed" should be "are closer" or "match better".**

Changed to "are closer". **L362**

**363: "most" should be "largest".**

Fixed; **L373**

**369-372: In this paragraph you could mention that only data from one cruise is used.**
**Otherwise on l. 372: during this cruise" the reader does not know which cruise is meant.**

We have adjusted this sentence to make it clear that the figure we present is from a single
cruise: "When compared against bottle data collected during a single hydrographic cruise
across Atlantic Repeat Hydrography Line 7 West …". **382-383**

**383: At this point I suggest to clarify that you refer to exchange with the interior basin/central**
**Labrador Sea.**
We have added "with the interior basin" to clarify that there is little exchange between the
Labrador Current and the interior Labrador Sea: "Little exchange with the interior basin appears
to occur along the Labrador Current until the vicinity of Flemish Cap …". **L397**

**698: It seems like the in Fig. 7, the titles for (c) and (d) are wrong. The upper panel should be**
**Ring A and the lower panel Ring B.**
Correct, thanks for catching this. We have fixed the titles for Figure 7 for Ring A and B. **L713**

[revised manuscript text omitted]